# Trading off Utility, Informativeness, and Complexity in Emergent Communication

**Mycal Tucker**
M.I.T.
mycal@mit.edu

**Roger Levy**
M.I.T.
rplevy@mit.edu

**Julie Shah**
M.I.T.
julie_a_shah@csail.mit.edu

**Noga Zaslavsky**
M.I.T.
nogazs@mit.edu

## Abstract

Emergent communication (EC) research often focuses on optimizing task-specific utility as a driver for communication. However, there is increasing evidence that human languages are shaped by task-general communicative constraints and evolve under pressure to optimize the Information Bottleneck (IB) tradeoff between the informativeness and complexity of the lexicon. Here, we integrate these two approaches by trading off utility, informativeness, and complexity in EC. To this end, we propose Vector-Quantized Variational Information Bottleneck (VQ-VIB), a method for training neural agents to encode inputs into discrete signals embedded in a continuous space. We evaluate our approach in multi-agent reinforcement learning settings and in color reference games and show that: (1) VQ-VIB agents can continuously adapt to changing communicative needs and, in the color domain, align with human languages; (2) the emergent VQ-VIB embedding spaces are semantically meaningful and perceptually grounded; and (3) encouraging informativeness leads to faster convergence rates and improved utility, both in VQ-VIB and in prior neural architectures for symbolic EC, with VQ-VIB achieving higher utility for any given complexity. This work offers a new framework for EC that is grounded in information-theoretic principles that are believed to characterize human language evolution and that may facilitate human-agent interaction.

## 1 Introduction

Good communication is a critical component of successful teams of humans and artificial agents, but differing notions of "good" makes training agents to develop such communication challenging. One view of communication focuses on *utility* by framing languages as successful to the extent they enable high task performance. Emergent communication (EC) literature, wherein agents learn to communicate while optimizing the utility associated with a task-specific goal, such as reaching a target landmark, emphasizes this view [1–6]. However, training via utility alone may lead to undesirable properties of the EC system, such as violations of linguistic universals [7], poor human interpretability [8], slow convergence rates [9], or high sensitivity to environment design [10]. This implies that some constraints may be needed to guide agents toward human-like communication.

Another view of communication, often considered in the cognitive science literature, focuses on task-agnostic communicative objectives as major forces shaping human languages [11, 12]. Most relevant to our work is a recent line of work [13–17] that argues that languages evolve under pressure to efficiently compress meanings into communicative signals by optimizing the Information Bottleneck (IB) principle [18]. In this context, IB can be interpreted as a tradeoff between the

36th Conference on Neural Information Processing Systems (NeurIPS 2022).

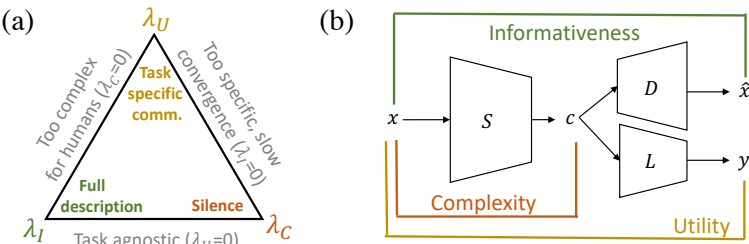

Figure 1: (a) In our theoretical framework, agents evolve to communicate with each other by optimizing a tradeoff between utility, informativeness, and complexity, weighted by $\lambda_U$, $\lambda_I$, and $\lambda_C$ respectively. (b) Basic setup and agent architecture. A speaker, $S$, maps an input, $x$, to a signal, $c$. A listener, $L$, takes an action, $y$, and reconstructs $x$ using a decoder $D$.

complexity and informativeness of the lexicon [13]. Intuitively, complexity corresponds to the number of bits allocated for communication, and informativeness corresponds to how well a listener can understand a speaker regardless of any task-specific goal (e.g., humans understand the term "blue" in "avoid the blue box" and "find a blue cup"). This information-theoretic framework for grounded semantic categories has been gaining broad empirical support across hundreds of languages and multiple semantic domains [13–16], and has been directly tied to language change over time [17]. However, the actual process that may drive a population of speakers — whether human or artificial — toward IB-efficient communication systems in a given environment remains largely unknown.

In this work, we integrate these two views of communication by training teams of interactive neural agents to optimize a tradeoff between maximizing utility, maximizing informativeness, and minimizing complexity, as depicted in Figure 1. We hypothesize that taking into account complexity and informativeness, which are cognitively-motivated task-general communicative facctors, in addition to task-specific utility, may help guide artificial agents toward communication systems that can be more naturally aligned with and adapted to human languages. To this end, building on prior work [19, 20], we propose Vector-Quantized Variational Information Bottleneck (VQ-VIB), a method for training neural agents to encode inputs into signals, while maintaining a discrete (symbolic) signal representation that is embedded in a continuous space, inspired by word embeddings [21–23]. We evaluate our approach in two multi-agent reinforcement learning (MARL) environments and in two variants of a color reference game. We show that: (1) by annealing a single tradeoff parameter, VQ-VIB agents can continuously adapt to changing communicative pressures and, in the color domain, align with human languages, without changing the agents' architecture or the characteristics of the training environment; (2) the emergent VQ-VIB embedding spaces capture semantically meaningful structures, grounded in the perceived environment; and (3) encouraging agents to be informative generally leads to faster convergence rates and improved utility, with VQ-VIB achieving higher utility for any given complexity compared to previous EC approaches that also support symbolic communication. Taken together, this work offers a new information-theoretic framework for EC that is grounded in general optimization principles that are believed to characterize human language evolution, and that has the potential of improving human-agent interaction.

## 2 Related work

While EC studies usually focuses on task utility, some studies also consider communicative constraints that are closely related to our measures of complexity and informativeness (defined in Section 3). For example, numerous works consider discretization, environmental noise, or penalty on large vocabulary sizes as proxies for reducing complexity [23–25]. In contrast, our notion of complexity is grounded in rate-distortion theory [26], and has been shown to capture the degree of semantic compression seen across languages [13]. Wang et al. [27] use the same IB complexity measure, but in different communicative settings that involve a centralized message processor. Other notions of linguistic complexity often focus on grammar, word forms, or compositionality [28–31]; these are beyond the scope of this work but offer promising directions for future work. Apart from complexity, some works consider auxiliary reconstruction losses in EC [9, 32]. In particular, Lin et al. [33] use an autoencoding loss which is closely related our notion of informativeness. However, they do not penalize complexity and note that their method may fall short in cases where "agents need to be more

selective of the information communicated." Lastly, there is a body of work on reinforcement learning with informational constraints similar to ours [e.g., 34–38], however not in the context of EC.

To evaluate our method, we consider particle-world environments that agents must navigate [3, 23], as well as reference games (related to Lewis signalling games [39]), both of which are popular types of EC environments. For reference games, we focus on the domain of color for several reasons: First, it has been a central test-case for cognitive theories of semantic categorization and language evolution [1, 40, 41]. Second, it enables comparison with the World Color Survey [WCS; 42], a dataset of color naming systems from 110 languages of non-industrialized societies. Third, it has been a focus of numerous studies in EC and machine learning more generally [1, 43–45, 10]. In particular, Chaabouni et al. [10] show that color signaling systems in artificial agents may reach the IB efficiency bound from [13] and capture some of the variation seen across human languages. However, to achieve this, they hand-craft varying characteristics of the training environment by selecting dissimilar distractor colors or by adding noise to the agents' communication. Rather than focusing on environment-based factors, we adopt a principled approach by directly optimizing agents with respect to a variational IB principle in a given environment.

## 3 Information-theoretic emergent communication

Here, we propose an information-theoretic EC framework that integrates task-general communicative objectives with task-specific utility maximization. We begin by reviewing the theoretical framework on which we build (Section 3.1.1) and the set of variational inference tools that we use in order to extend that framework to EC settings (Sections 3.1.2 and 3.1.3). We then define an integrative objective function for training EC agents (Section 3.2) and present Vector-Quantized Variational Information Bottleneck (VQ-VIB, Section 3.3), a method for training agents with respect to our objective function while maintaining structured symbolic representations that can be used for communication. The experimental setup and results for evaluating this new method are presented in Sections 4 and 5.

### 3.1 Preliminaries

#### 3.1.1 The Information Bottleneck framework for grounded semantic systems

We build on and extend Zaslavsky et al. [13]'s information-theoretic framework for grounded semantic systems. In this framework, speakers and listeners jointly optimize the IB tradeoff [18] between the complexity and informativeness of their communication system. A speaker is characterized as a probabilistic encoder $S(c|x)$ that, given an input $x \sim p(x)$, mentally represents it as $m_x$ and generates a communication signal, $c$. $m_x$ is the speaker's belief state, which is grounded in the agents' representation of the world and captures any uncertainly the speaker may have about the input. More formally, $m_x$ is defined by a probability distribution over a set of features $u \in \mathcal{U}$ that characterize the input space, i.e, $m_x(u) \equiv p(u|m_x)$. For example, $x$ may be a target color and $m_x$ a Gaussian distribution over the CIELAB perceptual color space (Figure 3c), capturing the speaker's perceptual uncertainly. Given $c$, the listener infers the speaker's belief state by performing Bayesian inference, i.e., it constructs an estimator $\hat{m}_c(u) = \sum_x S(x|c)m_x(u)$ using the posterior $S(x|c) \propto S(c|x)p(x)$.

Complexity is measured by the mutual information between the speaker's inputs and signals, $I_S(x, c)$, which, as in rate-distortion theory [26, 46], is roughly the number of bits that are needed for communication [18]. Informativeness corresponds to the similarity between the speaker's and listener's belief states. Because these states are defined by probability distributions, maximizing informativeness amounts to minimizing the expected Kullback-Leibler (KL) divergence between them, i.e., $\mathbb{E}_S[D_{\mathrm{KL}}[m_x \| \hat{m}_c]]$. Taken together, in this view, for any given tradeoff parameter $\beta \geq 0$ the speaker and listener should jointly minimize

$$I_S(x, c) + \beta \mathbb{E}_S[D_{\mathrm{KL}}[m_x \| \hat{m}_c]], \tag{1}$$

which is equivalent to optimizing the IB principle [see 13, 47, for derivation]. For $\beta \leq 1$, the optimal systems are non-informative and minimally complex ($I_S(x, c) = 0$), yielding no communication. Gradually annealing $\beta$ yields optimal systems with greater complexity, and as $\beta \to \infty$, the optimal systems become maximally informative but also maximally complex. Zaslavsky et al. [13] and colleagues have shown that human languages achieve near-optimal IB tradeoffs across multiple semantic domains (see Figure 6a for example), and that the continuous evolution of the optimal IB systems via annealing the tradeoff parameter $\beta$ captures important aspects of language evolution [13, 17].

### 3.1.2 Variational Information Bottleneck

Variational Information Bottleneck [VIB, 19] is a variational approximation method for IB with continuous representations. In its typical form, a neural encoder $q_\theta(z|x)$ maps an input $x$ to parameters of a $d$-dimensional Gaussian distribution, $\mu(x)$ and $\Sigma(x)$, from which a latent representation $z \in \mathbb{R}^d$ is sampled. Given $z$, VIB reconstructs a target feature $u$ using $q_\phi(u|z)$, parameterized by a soft-max function. VIB optimizes a variational bound on the IB objective function, based on the fact that

$$I_{q_\theta}(x; z) \le \mathbb{E}\left[D[q_\theta(z|x)\|r(z)]\right] \tag{2}$$

for any distribution $r(z)$, which is often taken to be $\mathcal{N}(0, I_d)$. VIB also bounds the IB informativeness term. However, because we consider here a different instantiation of IB, based on unsupervised rather than supervised learning, we use a different informativeness bound, as explained in Section 3.3.

### 3.1.3 Vector-Quantized Variational Autoencoder

While VIB learns continuous representations, Vector-Quantized Variational Autoencoder [VQ-VAE, 20] learns discrete, but potentially highly complex, representations in an unsupervised manner. More specifically, VQ-VAE defines a latent embedding space of $K$ vectors $\zeta_i \in \mathbb{R}^d, i = 1, \ldots, K$. A deterministic encoder maps an input $x$ to a discrete variable $i$ by first passing $x$ through a deterministic network to obtain a latent representation $z(x) \in \mathbb{R}^d$, and then discretizing it to the index of the nearest embedding vector, i.e., $i = \operatorname{argmin}_j \|z(x) - \zeta_j\|$. The decoder receives the corresponding embedding vector, $\zeta_i(x)$, and passes it through another deterministic network to reconstruct the input. The set of trainable parameters $\Theta$ consists of the embedding vectors and the parameters of the encoder and decoder networks. Because gradients cannot be passed through the `argmin` operator, a straight-through estimator [48] is used. Similar to VAE [49], VQ-VAE's training loss (Eq. (3)) uses the evidence lower bound (ELBO), which is reduced in this case to the first term in Eq. (3). VQ-VAE adds to it two VQ terms, designed for learning the embedding space, yielding the overall training loss

$$l_{\text{VQ-VAE}} = \log p(x|\zeta_i(x); \Theta) + \|\mathtt{sg}[z(x)] - \zeta_i(x)\|^2 + \alpha\|z(x) - \mathtt{sg}[\zeta_i(x)]\|^2, \tag{3}$$

where `sg` stands for the `stopgradient` operator, and following [20], $\alpha = 0.25$.

### 3.2 Integrating utility, informativeness, and complexity

The theoretical framework and variational tools reviewed in section 3.1 focus on general informational constraints and autoencoding reconstruction losses. However, EC settings are often based on multi-agent reinforcement learning (MARL), where agents are trained to maximize an expected reward, or utility, for a given task. In a basic cooperative setting, there are two agents: a speaker and a listener. The speaker receives an input, $x \in \mathcal{X}$, which is sampled from a distribution $p(x)$ and contains information about the task's goal which is hidden from the listener. The speaker's actions are restricted to communicating a vector $c \in \mathcal{C}$ to the listener. The listener receives $c$ and a partial observation $o \in \mathcal{O}$ about the world state, and takes an action $y \in \mathcal{Y}$. We denote the speaker's policy by $S(c|x)$ and the listener's policy by $L(y|c, o)$. In each episode the agents receive a utility $U(x, y)$ based on their actions and aim to learn policies that maximize the expected utility $\mathbb{E}_{S,L}[U(x, y)]$.

Note that this setting is similar to that of Section 3.1.1, with the key difference being that here the listener takes an action in the world instead of reconstructing the speaker's belief state. To integrate the two settings, we add to the basic EC setting a decoder, $D$, as shown in Figure 1b. If the speaker has no uncertainty about the input, then the decoder maps $c$ to a reconstruction $\hat{x}_c$. Otherwise, the decoder reconstructs a belief state, $\hat{m}_c$, as explained in Section 3.1.1. A similar approach can also be applied to more complex EC settings, which may involve sequential actions with discounted rewards, additional agents, or agents that can simultaneously communicate and act. In Sections 4 and 5, we consider navigation tasks with sequential actions, and reference games which correspond to the basic setting. We leave the exploration of other multi-agent settings to future work.

Our proposed objective function for training EC agents in these settings combines utility maximization with the IB objective from Eq. (1), yielding

$$\mathcal{L}_\lambda[S, L, D] = \lambda_U \mathbb{E}_{S,L}[U(x, y)] - \lambda_I \mathbb{E}_{S,D}[D_{\text{KL}}[m_x\|\hat{m}_c]] - \lambda_C I_S(x, c), \tag{4}$$

such that agents should jointly maximize $\mathcal{L}_\lambda[S, L, D]$. The non-negative tradeoff parameters, or Lagrange multiplies, $\lambda = (\lambda_U, \lambda_I, \lambda_C)$, control the relative weights of utility, informativeness

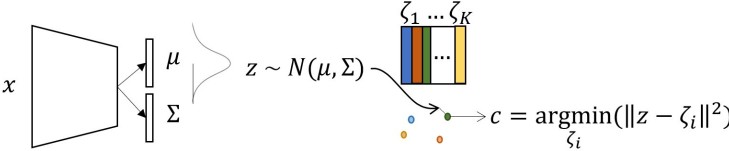

Figure 2: The VQ-VIB encoder architecture. The speaker encodes an input, $x$, as parameters to a Gaussian distribution, from which a vector is sampled and then discretized by mapping to the nearest encoding, to produce a communication vector, $c$.

(negatively related to the second term), and complexity. Intuitively, these three weights form a simplex (Figure 1a) that spans the range of communication systems that may emerge in our framework, from uninformative communication (minimizing complexity alone), to task-specific communication that may not generalize (maximizing utility alone), to complete descriptions (maximizing informativeness alone) that may be unnecessarily complex for many tasks and too complex for alignment with humans.

### 3.3 Vector-Quantized Variational Information Bottleneck

As in IB, directly maximizing Eq. (4) is intractable in large domains. To address this we take a variational approximation approach, where we assume parameterized agents and maximize a variational lower bound on Eq. (4). Additionally, we consider an encoder (speaker) architecture that learns symbolic representations that are embedded in a continuous space, inspired by word embeddings [22, and see also [23] for a related prototype-based EC approach]. We refer to this new method as Vector-Quantized Variational Information Bottleneck (VQ-VIB). Before defining our method precisely, we note that VQ-VIB builds on VIB and VQ-VAE, but at the same time differs from them in important ways. VQ-VIB learns a stochastic encoder, in contrast to VQ-VAE, and symbolic representations, in contrast to VIB. Additionally, while VQ-VIB's training objective borrows from VIB and VQ-VAE, it considers a different variational IB bound than VIB, uses that bound instead of the ELBO in VQ-VAE, and adds to that a symmetry-breaking term.[1] These difference are particularly crucial for human-like communication, as human languages employ symbolic representations (e.g., words), human speakers are often stochastic, and as discussed in Section 3.1.1, there is substantial empirical evidence that human languages are shaped by the IB principle.

#### 3.3.1 VQ-VIB architecture

The VQ-VIB encoder (illustrated in Figure 2) is composed of two components: a codebook of $K$ embedding vectors, $\mathcal{C} = \{\zeta_1, \ldots, \zeta_K\} \subset \mathbb{R}^d$, similar to VQ-VAE, and a VIB-like stochastic encoder $q_\theta(z|x)$. Given an input $x$, the encoder $q_\theta(z|x)$ is used to sample a latent representation $z \in \mathbb{R}^d$, followed by a quantization layer that discretizes $z$ into the nearest embedding vector. This encoding process generates a vector $c = \text{argmin}_{\zeta \in \mathcal{C}} \|z - \zeta\|^2$, which in our settings will be used for communication. We instantiate $q_\theta(z|x)$ by a feedforward network that maps $x$ to parameters of a Gaussian distribution over the latent space. Overall, the VQ-VIB encoder, or speaker, is given by

$$S_\theta(c|x) = \mathbb{P}(c = \underset{\zeta \in \mathcal{C}}{\text{argmin}} \|z - \zeta\|^2 | x, \mathcal{C}, \theta). \tag{5}$$

Finally, as shown in Figure 1b, we pair the VQ-VIB encoder with a simple decoder network $D_\psi(\hat{x}|c)$ that reconstructs the speaker's input given the communication vector (see Section 3.3.2 for justification), and with a task-specific policy network $L_\phi(y|c, o)$ from which the listener selects its actions. These two networks can also be paired with other speaker (encoder) architectures, as we do in subsequent sections in order to evaluate the VQ-VIB method.

#### 3.3.2 Learning objective

To train VQ-VIB agents, we first define a variational bound on Eq. (4). We bound the complexity term (third term in Eq. (4)) by noting that $I_{S_\theta}(x; c) \leq I_{q_\theta}(x; z) \leq \mathbb{E}[D_{\text{KL}}[q_\theta(z|x)\|r(z)]]$, where the first inequality folows from the data processing inequality, and the second one follows from Eq. (2).

---

[1]VIB and VQ-VAE also do not optimize the utility term in Eq. (4), but that can be easily added to them.

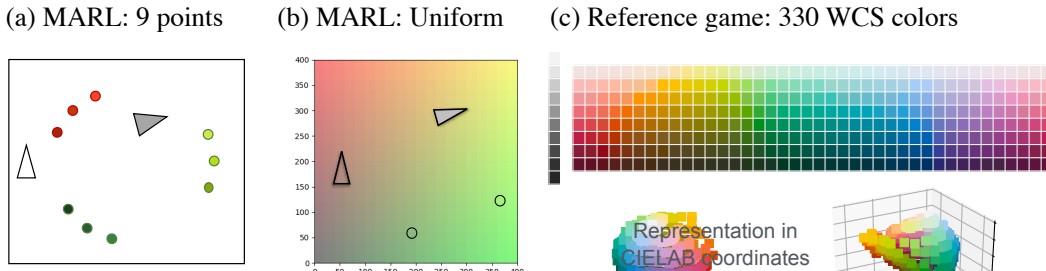

(a) MARL: 9 points (b) MARL: Uniform (c) Reference game: 330 WCS colors

Figure 3: Test environments. In the MARL settings (a-b) a speaker (clear triangle) observes a target location drawn either from a fixed set of 9 points (a) or uniformly in a continuous space (b), communicates it to a listener (gray triangle), which then needs to navigate to that location (adapted from [23]). In the color reference game (c), a speaker observes a target color from the World Color Survey [WCS, 42] palette, represented in the 3D CIELAB perceptual color space, and communicates it to the listener, who then needs to identify the target color.

To bound the informativeness loss (second term in Eq. (4)), we first assume that given $x \in \mathbb{R}^n$, the speaker's uncertainty is captured by Gaussian noise and therefore its belief states are given by $m_x = \mathcal{N}(x, \Sigma)$. Under this assumption, it holds that $\mathbb{E}[D_{\mathrm{KL}}[m_x \| \hat{m}_c]] \leq \frac{1}{2} \mathbb{E}[\|x - \hat{x}_c\|^2] + const$ (see Appendix A for proof), allowing us to approximate the ideal decoder by a simplified decoder $D_\psi : \mathcal{C} \to \mathbb{R}^n$ that outputs $\hat{x}_c$. This yields the following lower bound on Eq. (4):

$$\tilde{\mathcal{L}}_\lambda[\theta, \phi, \psi, \mathcal{C}] = \lambda_U \mathbb{E}_{S_\theta, L_\phi}[U(x, y)] - \lambda_I \mathbb{E}_{S_\theta, D_\psi}[\|x - \hat{x}_c\|^2] - \lambda_C \mathbb{E}_{q_\theta}[D_{\mathrm{KL}}[q_\theta(z|x) \| r(z)]]. \quad (6)$$

To train the embedding vectors, we include the two VQ terms from Eq. (3), and based on [50], we also add a small regularization term, $\varepsilon(\theta, \mathcal{C})$, designed to break the symmetry between equally good solutions (see Appendix B). Taken together, we obtain the following objective function for VQ-VIB:

$$\mathcal{L}_{\text{VQ-VIB-}\lambda}[\theta, \phi, \psi, \mathcal{C}] = \tilde{\mathcal{L}}_\lambda[\theta, \phi, \psi, \mathcal{C}] - \mathbb{E}_{S_\theta} \left[ \|\mathtt{sg}[z] - c\|^2 + \alpha \|z - \mathtt{sg}[c]\|^2 \right] - \varepsilon(\theta, \mathcal{C}), \quad (7)$$

which should be jointly maximized by the agents for any given $\lambda$. Following [19], we fix $r(z) = \mathcal{N}(0, I_d)$, and following [20], we fix $\alpha = 0.25$. To optimize this objective function, we use the VQ-VAE straight-through estimator to bypass the non-differentiablity of the $\mathtt{argmin}$ operator, and the reparameterization trick [49] for backpropagating through $z$. While the latter is internal to the speaker, the straight-through estimator is used to pass gradients from the decoder $D_\psi$ and the listener's policy $L_\phi$ into the speaker $S_\theta$. Here, we allow gradients to pass from the decoder, but take a more careful approach to passing gradients from the listener by comparing the straight-through estimator approach to methods that do not pass gradients, as explained in the next section.

## 4 Experimental setup

To evaluate the emergent VQ-VIB communication systems and how they are shaped by $\lambda$, we consider three test environments (Figure 3) and compare VQ-VIB with strong baselines, as described below. Full details of the architectures, hyperparameters, and environments are provided in Appendix D.[2]

**Trials and baselines** We consider three types of baseline speaker architectures that are derived from prior work: (1) a continuous speaker [e.g., 51], which unlike humans, communicates using continuous vectors without maintaining symbolic representations; (2) a one-hot speaker [e.g., 3], which communicates with discrete but unstructured vectors; and (3) a prototype-based speaker [Proto, 23], which maps one-hot vectors to an embedding space, and in that sense is similar to VQ-VIB, although its internal mechanism is different. We also considered a variant of the VQ-VIB speaker ('VQ-VIB After') which is more similar to Proto, but because it does not perform as well as VQ-VIB we leave its discussion to Appendix C. For a fair comparison, we train all speakers with the same decoder and listener architectures, allow them to be stochastic similar to the VQ-VIB speaker, and

---

[2]Code is available at https://github.com/mycal-tucker/vqvib_neurips2022

adjusted their training objective according to Eq. (6) to allow them to consider the same informational constraints as VQ-VIB. In all domains, agents are trained from scratch using 5 random seeds.

**Particle worlds** We train agents in the two particle world environments shown in Figure 3, which are adapted from Tucker et al. [23]. Both environments consist of two-dimensional worlds in which a speaker observes the exact target location that a listener needs to reach but cannot directly observe. In the 9 points environment, there are 9 landmarks at fixed locations, one of which is chosen at random as the target for each episode. In the Uniform environment, two landmarks are spawned in randomly drawn locations in the continuous space and one of them is specified as the target. Both agents receive a reward equal to the negative distance from the listener to the target and are trained using Multi-Agent Deep Deterministic Policy Gradient [3], which is a standard policy-gradient method without backpropagating from one agent into another. The listener cannot observe the target location, and the speaker can only communicate at the first timestep by broadcasting a vector that is observable at all future timesteps; thus, an optimal team policy consists of the speaker communicating about the target at the first timestep, and the listener navigating to the target based on the message.

**Color reference games** We consider two variants of a color reference game, a basic game and a common ground game (see Figure D.1). In the basic game, a speaker observes a target color and outputs a communication vector. A listener observes the speaker's communication vector and two colors — the target and a distractor, without knowing which is the target — and tries to guess the target. In the common ground game, the speaker also observes the distractor, in addition to the target, which in principle allows it to communicate in a more pragmatic way by taking into account how the distractor may influence the listener's choices [44, 52]. Apart from that, the two variants are the same. Following [41, 13], colors are represented in the 3D CIELAB color space (Figure 3c), in which small Euclidean distances between colors are correlated with human perceptual dissimilarities. To capture perceptual noise, the speaker's observations are corrupted by $\epsilon \sim \mathcal{N}(0, \sigma^2 I)$ with $\sigma^2 = 64$, as motivated by human perception [53]. The target and distractor are sampled independently from the 330 color chips of the World Color Survey [WCS, 42] shown in Figure 3c, according to a human-based prior from [13].[3] Agents are trained with respect to the binary crossentropy loss computed on the listener's output. We consider two training methods: (i) using straight-through gradient estimators to backpropogate from the listener into the speaker, which is common in EC and supports efficient training, although it is not biologically plausible; and (ii) using the REINFORCE algorithm [54], which does not pass gradients between agents but is known to suffer from slow convergence rates.

## 5 Results

### 5.1 Particle worlds

We begin with exploring the influence of $\lambda$ on the emergent agent behavior in the two particle world environments. Without loss of generality, we take $\lambda_U = 1$ unless stated otherwise, and then evaluate the influence of $\lambda_C$ and $\lambda_I$ separately. Starting with $\lambda_C$, we simulate the evolution of the emergent VQ-VIB communication systems by fixing $\lambda_I = 1$ and varying $\lambda_C$ using deterministic annealing [55] (see Appendix D for detail), much like the evolution of the IB representations in [50]. Figure 4 shows lower- and higher-complexity solutions for the two environments (see Movie 1 for the continuous evolution of the systems). The top panels (Figure 4a and c) show how the speaker partitions the environment into communication signals, together with the listener's trajectories in response to the speaker's communication. The lower panels (Figure 4b and d) show the emergent VQ-VIB embedding space $\mathcal{C}$. In both environments, lower complexity solutions form a coarser representation of the space, with fewer communication vectors (signals) used by the speaker and, respectively, fewer final positions reached by the listener. We wish to emphasize that the number of signals available to the speaker is fixed throughout these experiments ($K = 9$ in the 9 points environment; $K = 100$ in the uniform environment), as well as the agents' architectures and all environment parameters. The structural changes in the *effective* number of signals used by the speaker result from annealing $\lambda_C$, which dynamically controls the communicative bandwidth that the agents coordinate during self-play.

Remarkably, the emergent structure of the VQ-VIB embedding spaces (Figure 4b and d) captures semantically meaningful information that is grounded in the environment. That is, the embedding vectors roughly encode distances between "named" regions. We note that Tucker et al. [23]'s

---

[3]The WCS data are available at `www1.icsi.berkeley.edu/wcs` and the human-based color prior is available at `https://github.com/nogazs/ib-color-naming`.

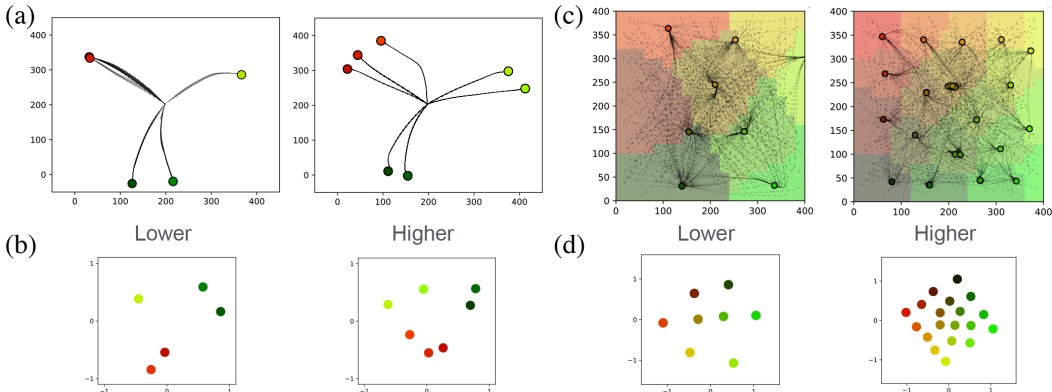

Figure 4: VQ-VIB solutions for the 9 points (a-b) and uniform (c-d) environments, with lower (left panels) and higher (right panels) complexity (see also Movie 1). Each communication vector is associated with the mean color of the positions it can refer to, based on the speaker's encoder and the colors in Figure 3. (a) Listener's trajectories (black curves) and final positions (dots) in response to a speaker's communication vector, indicated by the color of the final dot. (b) The emergent VQ-VIB communication vectors (trained with $d = 2$). (c) Same as (a) but for the uniform environment. Background colors show the modal communication vector for each target position in the continuous space. (d) 2D PCA of the VQ-VIB communication vectors (trained with $d = 3$).

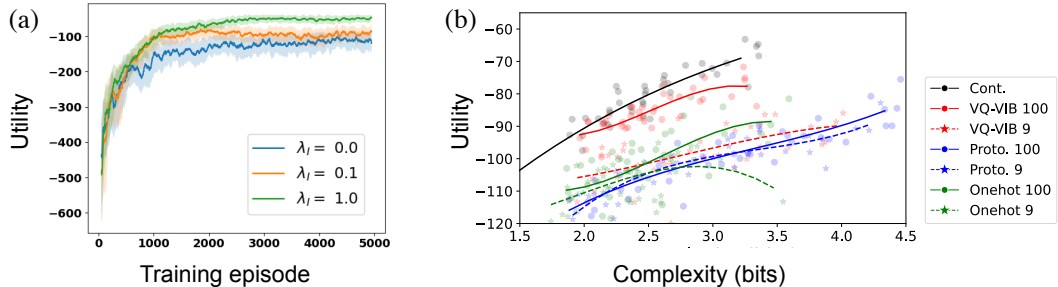

Figure 5: Uniform environment results. (a) Expected team utility during training of VQ-VIB agents with varying informativeness pressures. (b) Utility vs.complexity for the continuous architecture and the symbolic EC architectures with $K = 9$ (dashed curves and stars) and $K = 100$ (solid curves and dots) signals, excluding a few points below -120 for visualization only. Each curve corresponds to the cubic best fit to the data (stars/dots) generated across 5 random seeds for each configuration.

prototype-based approach is also capable of learning such grounded representations, although without informational constraints, thus resulting in maximally complex and task-specific communication. Therefore, as explained in Section 4, we compare VQ-VIB with an adjusted version of Tucker et al. [23]'s approach (Proto), in addition to a one-hot and a continuous architecture. While the latter is not human-like, we consider it here as an upper bound on the achievable performance in these environments. The results for the uniform environment are shown in Figure 5b, and similar trends also hold in the 9 points environment (Appendix E). It can be seen that VQ-VIB achieves higher utility for any given complexity and codebook size $K$ compared to the two baselines that support discrete (symbolic) communication, i.e., Proto and one-hot. Additionally, increasing $K$ from 9 to 100 substantially improves VQ-VIB, bringing it close to the performance of the continuous architecture, while the same increase in $K$ yields much smaller improvements for Proto and one-hot.

Next, we evaluate the influence of the informativeness weight, $\lambda_I$. While informativeness and utility may seem similar, and they are indeed closely related, they are also importantly different: maximizing utility encourages agents to coordinate on a task-specific communication system, whereas maximizing informativeness encourages agents to develop a shared understanding regardless of any task-specific needs. Therefore, intuitively, informativeness may help agents coordinate faster and converge to more effective communication systems. To test this intuition, we eliminate the complexity constraint by setting $\lambda_C = 0$ and vary $\lambda_I$. Figure 5a shows that, indeed, increasing $\lambda_I$ leads to higher utility and

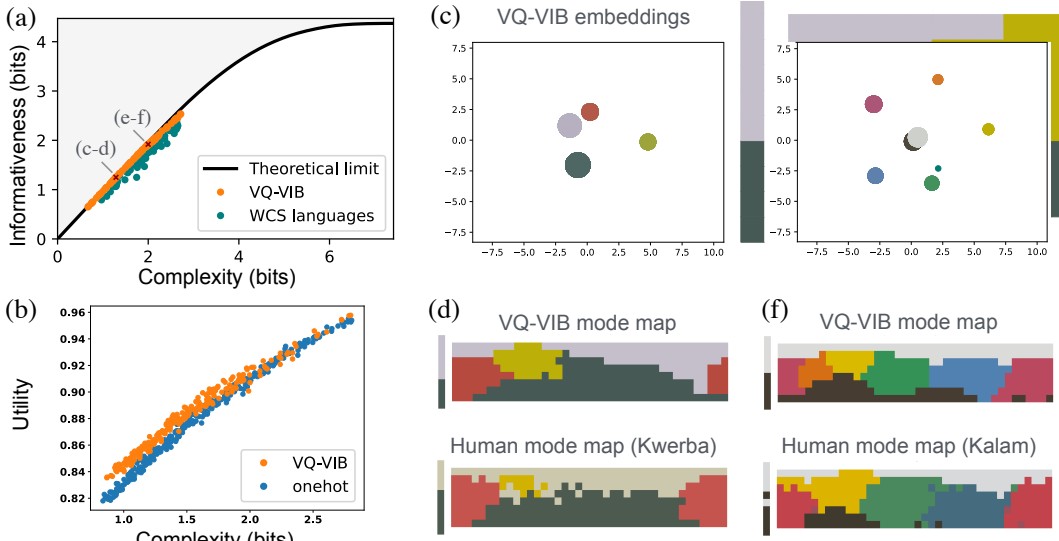

Figure 6: Color reference game results (basic setting, see also Movie 2). **(a)** VQ-VIB agents (orange dots) reach the theoretical bound from [13] (black curve) and capture the variation observed across the WCS languages (teal dots). **(b)** VQ-VIB achieves greater utility, for the same complexity, than one-hot. **(c-f)** The emergent VQ-VIB color naming systems and closest (min gNID) human systems in the WCS data, for agents with lower (c-d) and higher (e-f) complexity, as shown in (a). Each communication vector is associated with the mean color it can refer to. **(c, e)** 2D PCA of the VQ-VIB communication vectors, where the size of each point reflects its relative category size. **(d, f)** Mode maps, where each color chip from Figure 3c is colored according to its modal communication vector.

faster convergence rates. The results shown here are only for VQ-VIB in the uniform environment, but similar trends appear for all architectures in both environments (Appendix E).

### 5.2 Color reference games

Our results for the two types of color reference games (basic and common ground), and for the two types of training methods (REINFORCE and straight-through estimator) are similar, with REINFORCE being slightly worse (consistent with [10]). We therefore focus here on the basic game with straight-through estimator and provide the full set of results in Appendix F. In addition, because we wish to compare our agents' communication systems with actual human color naming systems, we focus only on VQ-VIB and one-hot. The continuous architecture cannot generate discrete color categories, and based on the previous section, Proto performs similarly to one-hot.

As before, we fix $\lambda_I = 1$ and train agents while annealing $\lambda_C$. We set $K = 330$ (the number of color chips) for all agents, which allows full expressivity, although that would require maximal complexity and is unlike attested human languages which typically use only a few color terms [40]. We therefore focus on the range of $\lambda_C$ that produces the variation in complexity observed across languages. Following Zaslavsky et al. [13], we use the color naming data from the World Color Survey [WCS, 42] to estimate an encoder $S_l(c|x)$ for each WCS language $l$. Figure 6a replicates Zaslavsky et al. [13]'s finding that languages are near-optimal with respect to the IB bound, and further shows that VQ-VIB agents can also achieve that bound via self-play. Visualizing the emergent VQ-VIB communication systems (Figure 6c-f) shows, as before, the structural changes in the effective codebook size as complexity increases (see Movie 2 for the continuous evolution of these systems), as well as the similarity between the VQ-VIB speaker color-to-signal mapping and the nearest WCS language (Figure 6d and f). This recapitulates the theoretical evolutionary process derived by [13], while additionally taking into account local agent interactions and a learned embedding space.

This embedding space is particularly noteworthy, as it is grounded in human color perception and captures semantically meaningful information about the perceptual color space. At low complexity (Figure 6c), the embeddings reflect the irregularity of the CIELAB space due to the strong perceptual

Table 1: Quantitative evaluation of the color communication systems of annealed agents (VQ-VIB, One-hot), utility-only VQ-VIB agents, and the IB-optimal systems from [13]. Human-agent gNID is the average dissimilarity ($\pm$STD) between human color naming systems and nearest agent system. Agent efficiency loss is the average deviation from optimality ($\pm$STD) of the agents' systems, measured with respect to the theoretical bound from [13]. For both measures, lower values are better.

|  | VQ-VIB | One-hot | VQ-VIB (utility-only) | IB-optimal |
|---|---|---|---|---|
| Human-agent gNID | **0.16** (0.01) | **0.16** (0.00) | 0.42 (0.02) | 0.18 (0.10) |
| Efficiency loss | **0.03** (0.00) | 0.05 (0.00) | 0.11 (0.01) | N/A, 0 by definition |

salience of yellow (Figure 3c). At higher complexity (Figure 6e), the embeddings form a hue circle, with achromatic colors (black and white) in the middle, as in CIELAB. Intriguingly, Shepard and Cooper [56] found that human similarity judgments for color words also recapitulate the hue circle, suggesting that VQ-VIB may potentially provide a possible mechanistic explanation for the emergence of grounded semantic representations both in humans and in artificial agents.

Finally, we measure the agents' deviation from the IB bound (efficiency loss, as in [13]) and their structural similarity to the nearest WCS languages (using gNID [13], a dissimilarity measure for encoders) for both VQ-VIB and one-hot agents. Table 1 shows that both VQ-VIB and one-hot agents learn human-like systems across the complexity spectrum, confirming the patterns in Figure 6. In addition, both are slightly better (lower human-agent gNID) compared to the IB-optimal systems from [13] that do not take into account utility. It is not surprising that both a stochastic one-hot encoder and a VQ-VIB encoder can perform well within the IB framework, because the traditional IB formulation with discrete representations does not assume any typology on the representation space. However, when assessed with respect to utility (Figure 6b), VQ-VIB outperforms the one-hot architecture, as in the particle world domains, suggesting that an embedding space may be particularly useful for downstream decision making. Training only to maximize utility ($\lambda_I = \lambda_C = 0$), however, causes VQ-VIB agents to converge to a narrow range of communication that fails to capture the diversity of human languages (Table 1, VQ-VIB (utility-only); training utility-only one-hot agents led to similar findings; see Appendix F). This suggests that neither communicative efficiency alone, at least as captured by IB, nor utility alone, but rather their combination may be important for the emergence of human-like communication.

## 6 Conclusions

In this work, we have proposed an information-theoretic framework for emergent communication that integrates two views: task-specific utility maximization, and general communicative constraints derived from the Information Bottleneck (IB) principle [18, 13]. To train agents within this framework, we have introduce a new method, VQ-VIB, that builds on VIB [19] and VQ-VAE [20], but also differs from them in ways that are particularly important for modeling human languages (Section 3.3). We have shown, across three types of test environments, that VQ-VIB agents can continuously evolve and adapt to changing communicative pressures via annealing a single tradeoff parameter, while learning grounded embedding spaces for communication that capture semantically meaningful information about the perceived environment, reminiscent of word embeddings. VQ-VIB also outperforms prior methods that support discrete (symbolic) communication, and in the domain of color, it can be aligned with human languages and possibly also with human similarity judgments. Natural extensions of our work include more complex domains and challenging tasks, as well as human-agent interaction experiments, which may further demonstrate advantages of VQ-VIB. More broadly, we believe our framework could be useful for further developing and improving both emergent communication in artificial agents and models of human language evolution.

## Acknowledgments and Disclosure of Funding

M.T. was supported by an Amazon Science Hub Fellowship. N.Z. was supported by a K. Lisa Yang Integrative Computational Neuroscience Postdoctoral Fellowship. We thank Rahma Chaabouni for useful discussions on their prior work, and our anonymous reviewers for their helpful feedback.

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
