# Supplementary Material for
# Trading off Utility, Informativeness, and Complexity in Emergent Communication*

**Mycal Tucker**   **Roger Levy**   **Julie Shah**   **Noga Zaslavsky**

M.I.T.

## A  Variational bound on the informativeness loss

Here we prove the variational bound on the informativeness loss term (second term in Eq. (4)) that is used in Section 3.3.2 to derive the VQ-VIB learning objective. Recall that the speaker's belief states, $m_x$, are defined as probability distributions over a feature space $\mathcal{U}$ representing the domain $\mathcal{X}$. A decoder maps a communication signal $c \in \mathcal{C}$ to a reconstruction $\hat{m}_c$ of the speaker's belief. As shown in [1, SI Section 1.2], for any given speaker $S(c|x)$, the optimal decoder that minimizes $\mathbb{E}_S[D_{\mathrm{KL}}[m_x\|\hat{m}_c]]$ over all possible reconstructions $\hat{m}_c$ is Bayesian; that is, its reconstruction is given by $\hat{m}_c(u) = \sum_x S(x|c)m_x(u)$, where $S(x|c)$ is the posterior distribution with respect to $S(c|x)$ and $p(x)$. Recall that $\hat{m}_c(u)$ is exactly the listener's decoder in the IB framework (see Section 3.1.1). Therefore, any other decoder would lend an upper bound on the informativeness loss term. Here, we derive a particularly useful bound under the assumptions that $\mathcal{X} \subseteq \mathbb{R}^n$,[2] $\mathcal{U} = \mathbb{R}^n$, and that the speaker's uncertainty can be modeled with Gaussian noise, $\mathcal{N}(0, \Sigma)$. For simplicity, we assume that the noise covariance, $\Sigma$, does not depend on the input, however this assumption is not required for our general claim and could be easily relaxed. In this case, the speaker's belief states are given by $m_x \equiv \mathcal{N}(x, \Sigma)$. Let $D_\psi : \mathcal{C} \to \mathbb{R}^d$ be a function parameterized by $\psi$, then it holds that

$$\min_{\hat{m}'_c} \mathbb{E}_S[D_{\mathrm{KL}}[m_x\|\hat{m}'_c]] = \mathbb{E}_S[D_{\mathrm{KL}}[\mathcal{N}(x, \Sigma)\|\hat{m}_c]] \tag{A.1}$$

$$\leq \min_{\hat{\mu}(c),\hat{\Sigma}(c)} \mathbb{E}_S[D_{\mathrm{KL}}[\mathcal{N}(x, \Sigma)\|\mathcal{N}(\hat{\mu}(c), \hat{\Sigma}(c))]] \tag{A.2}$$

$$\leq \min_{\psi,\hat{\Sigma}(c)} \mathbb{E}_S[D_{\mathrm{KL}}[\mathcal{N}(x, \Sigma)\|\mathcal{N}(D_\psi(c), \hat{\Sigma}(c))]]. \tag{A.3}$$

While Eq. (A.1) follows from [1] and the inequality in Eq. (A.2) is straightforward, it has an interesting cognitive interpretation. Notice that under our assumptions, $\hat{m}_c$ is a Gaussian mixture, whereas the speaker's beliefs are simply Gaussian. Eq. (A.2) is equivalent to assuming that the listener's reconstructions are restricted to Gaussian distributions, similar to the speaker. Therefore, while this might not give a tight bound on the optimal IB solution, it may actually be more cognitively plausible. In addition, the optimal $\hat{\mu}(c)$ for Eq. (A.2) are given by the posterior centroids, i.e,

$$\hat{\mu}^*(c) = \mathbb{E}_S[x|c] = \sum_{x \in \mathcal{X}} S(x|c)x, \tag{A.4}$$

which means that the optimal $D_\psi^*$ for Eq. (A.3) will approximate these centroids. Eq. (A.3) has a convenient form as minimizing the KL divergence between two Gaussians,

$$\min_{\psi,\hat{\Sigma}(c)} \frac{1}{2}\mathbb{E}_S\left[\left(\mathrm{tr}(\hat{\Sigma}(c)\Sigma^{-1}) + (x - D_\psi(c))^\top \hat{\Sigma}(c)^{-1}(x - D_\psi(c)) - n + \ln\frac{|\hat{\Sigma}(c)|}{|\Sigma|}\right)\right], \tag{A.5}$$

---

*The main paper is available at https://openreview.net/pdf?id=O5arhQvBdH.

[2]While $\mathcal{X}$ may be a continuous space, in our settings it is either a finite dataset or it is approximated by a finite sample during training. Therefore, we treat it here as a discrete set.

36th Conference on Neural Information Processing Systems (NeurIPS 2022).

which in principle, could be used in our framework with a proper parameterization of $\hat{\Sigma}(c)$. However, for simplicity, we take $\hat{\Sigma}(c) = I_d$, which, in combination with Eqs. (A.1)-(A.3), yields the following bound:

$$\mathbb{E}_S[D_{\mathrm{KL}}[m_x|\hat{m}_c]] \leq \min_{\psi} \mathbb{E}_S[D_{\mathrm{KL}}[\mathcal{N}(x,\Sigma)\|\mathcal{N}(D_\psi(c), I_d]] \tag{A.6}$$

$$\leq \frac{1}{2}\mathbb{E}_S\left[\|x - D_\psi(c)\|^2\right] + a\,, \tag{A.7}$$

where $a = \mathrm{tr}(\Sigma^{-1}) - n - \ln|\Sigma|$ is a constant that does not affect the optimization.

## B  Symmetry breaking regularization

As shown in [2], for every complexity level there is a set of IB-optimal systems with varying *effective* lexicon sizes, $k = |\{c \in \mathcal{C} : \sum_x p(x)S_\theta(c|x) > 0\}|$. All the systems with the same $k$ form an equivalence class and the canonical system within each class is the one with minimal $k$. These canonical systems are the natural one to prefer, because they can attain the optimum for a given complexity with a minimal codebook. We therefore aim to bias our agents toward these systems. One way of achieving that is by regularizing the entropy of the speaker's communication vectors. However, this is problematic in our settings given the non-differentiablity of the `argmin` operation used in the VQ-VIB discretization layer. Instead, we use the entropy of an approximated categorical distribution based on the euclidean distance to each quantized vector, i.e., $p_{\mathrm{cat}}(\zeta_i|z) \propto e^{-(z-\zeta_i)^2}$ Because this entropy term is only used to break the symmetry between equally optimal systems, we use a small regularizing weight (0.05 in our experiments). We denote the weighted regularization term by $\varepsilon(\theta, \mathcal{C})$, as it depends both on the encoder $q_\theta(z|x)$ and on the embedding space $\mathcal{C}$.

## C  VQ-VIB After

We proposed an additional architecture, VQ-VIB After, that is closely related to our main VQ-VIB method. In VQ-VIB After, an encoder to generates a discrete representation $\zeta$, whereupon a communication vector $c$ is sampled from a Gaussian in the continuous space centered around $\zeta$. This "sample-after" method is depicted in Figure 8, and corresponds to an intuitive notion of choosing a word (a discrete embedding) deterministically but then being uncertain about the form (by sampling in the continuous space around it).

This architecture fits into the standard Information Bottleneck framework, as complexity can be penalized via standard variational methods. However, it is distinct from discrete communication methods because communication vectors are ultimately sampled from a continuous space. Thus, this work does not fit naturally into the main focus of this work, but we believe it could have applications in other settings, such as speech processing (wherein discrete words are pronounced in a continuous auditory space).

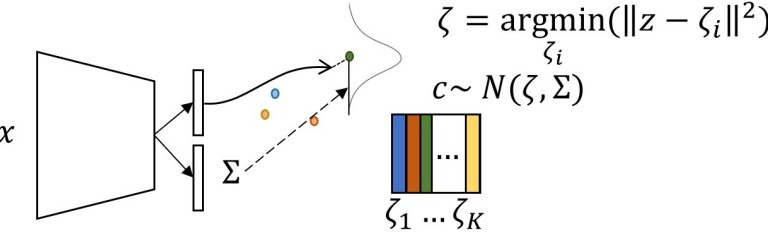

Figure C.1: In the VQ-VIB After architecture, the speaker deterministically maps $x$ to the nearest quantized encoding, $\zeta$, and then $c$ is sampled from a Gaussian centered at $\zeta$.

# D   Implementation Details

Here, we include implementation details that we omitted from the main paper that can be used to reproduce our experiments.

## D.1   Multi-Agent Reinforcement Learning Experiments

We trained agents in the 9 points and Uniform environments with Multi-Agent Deep Deterministic Policy Gradient (MADDPG), a standard method for multi-agent reinforcement learning [3].

Agent architectures were identical in both environments. Regardless of the speaker architecture used, all teams used common decoder and listener agents (except for variations due to different communication dimensions, explained later). The decoder used 2 fully-connected ReLU layers with hidden dimension 64; the listener used 3 fully-connected ReLU layers with hidden dimension 64.

Because of the common architecture for the decoder and listener agents across teams, variation arose due to differences among the speaker agents. A speaker agent consisted of 2, 64-neuron fully-connected layers with ReLU activations, followed by a speaker "head." We dub the output of these first two layers the "encoder representation." Speaker heads implemented the specific communication method (e.g., continuous, VQ-VIB, etc.) by taking the 64-dimensional encoder representation and going through one of 5 possible architectures to produce a vector in the communication space, $\mathbb{R}^d$. (As discussed later, $d$ varied by environment and architecture slightly.)

- **Continuous:** The continuous communication head consisted of two fully connected layers branching off the encoder representation to generate a $\mu$ and `logvar` (the log of the variance) from which the communication vector was sampled using the reparametrization trick. Here, as in other speaker architectures, we assumed a diagonal covariance matrix. Complexity was measured as the KL divergence between the parameters $\mu$ and `logvar` and a unit Gaussian.

- **one-hot:** The one-hot communication head consisted of one fully connected layer from the encoder representation to $\mathbb{R}^d$. One-hot vectors were generated using gumbel softmax sampling, with temperature 1. Complexity was measured as the KL divergence between the softmax distribution from which the one-hot vector was sampled and a uniform categorical distribution. Thus, complexity corresponds to the difference between the conditional and unconditional entropies of messages, unlike non-IB frameworks that often penalize unconditional entropy. We used greater communication dimensionality for one-hot agents and therefore rescaled the first layer of the decoder and listener agents to accept such communication.

- **VQ-VIB:** The VQ-VIB communication head first passed the encoder representation through two fully connected layers, branching off the encoding, to generate a $\mu$ and `logvar`, as in the continuous communication case. A communication vector was sampled from the distribution, whereupon it was discretized to the nearest quantized vector, $\zeta$, in the communication space (as done in a standard VQ-VAE, without the sampling). Complexity was measured as the KL divergence between the parameters $\mu$ and `logvar` and a unit Gaussian.

- **VQ-VIB After:** The VQ-VIB After communication head first passed the encoder representation through two fully connected layers to generate an encoding, $z$, and `logvar`. The encoding, $z$, was discretized to the nearest quantized vector, $\zeta$. Lastly, a communication vector, $c$, was sampled from a normal distribution centered at $\zeta$ with variance specified via `logvar`. Complexity was measured as the KL divergence between the parameters $\zeta$ and `logvar` and a unit Gaussian.

- **Proto:** The Proto head mapped the encoder representation through a single fully-connected layer to a categorical distribution over $k$ neurons, for $k$ representing the number of prototypes. A $k$-dimensional one-hot vector was sampled using gumbel softmax; a prototype, $p$ was selected by multiplying the prototype matrix by the one-hot vector. A different fully-connected layer mapped from the encoder representation to `logvar`. Lastly, a communication vector, $c$, was sampled from a Gaussian centered at $p$ with variance via `logvar`. Complexity was measured as the KL divergence between the parameters $p$ and `logvar` and a unit Gaussian. This closely resembles the VQ-VIB After architecture, but with a different discretization mechanism.

All agents were trained with an Adam optimizer with learning rate 0.01 and batch size 1024.

### D.1.1   9 points Environment

For all agent architectures except for one-hot, the communication dimension was 2; one-hot used communication dimension 9 so that the number of discrete representations allowed by the architecture was not the limiting factor for complexity. Similarly, the VQ-VIB and Proto agents used 9 discrete embeddings.

In our results with annealed complexity weight (presented in Figure E.3), agents were trained for 10,000 episodes, with each episode lasting 100 timesteps. The listener agent was always spawned at the origin, equidistant from each landmark. The informativeness loss weight, $\lambda_I$, was constant at 1.0. For the continuous and VQ-VIB communication methods, the complexity loss weight, $\lambda_C$ increased by $8 * 10^{-6}$ per episode. The one-hot and Proto methods suffered from communication collapse at that rate, so results $\lambda_C$ increased half as quickly: at $4 * 10^{-6}$ per episode.

Training a team in the 9 points environment for 10,000 episodes took approximately 15 minutes on a desktop computer with 16 Intel i9 cores for all architectures, except for one-hot, which took roughly 20 minutes. This difference in time is attributable to the larger communication dimension needed for one-hot.

### D.1.2   Uniform Environment

In the Uniform environment, we used identical parameters as in the 9 points environment, except for the changes noted below.

For all agent architectures except for one-hot, the communication dimension was 3; for one-hot, we experimented with communication dimensions of 9 and 100. Similarly, we tested VQ-VIB and Proto agents when using 9 or 100 discrete embeddings. The listener agent was spawned at a starting location sampled uniformly at random from the map (bounded between -200 and 200 for both $x$ and $y$). Otherwise, we used the same number of training episodes, episode duration, and complexity annealing rates as in 9 points.

Training a team in the Uniform environment for 10,000 episodes took approximately 15 minutes, except for one-hot, which took roughly 30 minutes for 100-dimensional communication.

### D.2   Color Reference Game Experiments

Here, we include the implementation details used in our color reference game experiments.

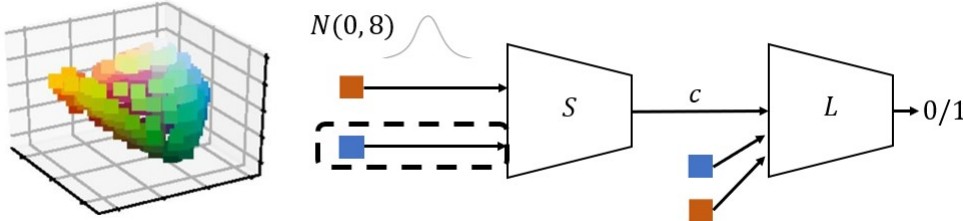

Figure D.1: Target and distractor colors were drawn uniformly without replacement from the set of 330 colors in CIELAB space (left). A speaker, $S$, observed the target and (in the Common Ground setting) distractor, corrupted by Gaussian noise. The listener, $L$, had to identify which of the shuffled colors it saw was the target color, given the speaker's communication, $c$.

Agents were trained in the overall architecture depicted in Figure D.1. The speaker agent was parametrized with 3 fully connected layers with ReLU activations and hidden dimension 64, followed by a communication head. The communication heads used the same architectures as specified in the particle world experiments: gumbel softmax was used for generating one-hot vectors, and a straight-through estimator was used for discretization in VQ-VIB architectures [4, 5]. In all experiments, the communication dimension was set to 32 for all agents except one-hot, for which the communication dimension was set to 330 (to match the expressive power of other discrete agents, that each had

330 discrete tokens in the 32 dimensional space). The decoder and listener agents used the same feed-forward architecture as in the particle world experiments.

Without annealing the complexity loss, 50 training epochs sufficed for agents to converge to a stable solution. Thus, in our experiments with $\lambda_C = 0$, we only trained for 50 epochs. For later experiments, wherein we annealed complexity, we first trained agents for 50 epochs before beginning to anneal the complexity term for the next 40 epochs (for a total of 90 epochs). We note that, for VQ-VIB, we set the complexity loss to 0 during this "burnin" period, but for one-hot communication, it appeared necessary to set the loss to 0.05 to prevent numerical instability during the first epoch after the burnin period. During complexity loss annealing, we increased $\lambda_C$ by 0.01 during each epoch for VQ-VIB, and by 0.1 for one-hot. These different rates were necessary to induce similar ranges in complexity of learned communication.

In the Basic color reference game, the speaker observed a only the 3-dimensional, corrupted target color. In the Common Ground setting, the speaker observed a 6-dimensional vector, generated by concatenating noisy versions of the target and distractor colors. In both settings, the listener observed the communication output by the speaker, as well as the target and distractor colors (ordered randomly, without added noise). Results were analyzed using code partially-based on publicly-available code released by Zaslavsky et al. [1].

We trained agents on a desktop computer with 16 Intel i9 CPUs and 1 NVIDIA GeForce RTX 2080. Training for 90 epochs in both the Basic and Common Ground settings took approximately 3 minutes for all agent architectures. Agents were trained with an Adam optimizer with default parameters, with batch size 1024.

# E   Full Particle World Results

For brevity, we only included partial results from the particle-world experiments in the main paper. Here, we include the full results.

First, we experimented with varying $\lambda_I$ for different environments and neural architectures, without annealing complexity. Training curves for both environments, using varying $\lambda_I$ for each neural architecture, are included in Figures E.1 and E.2. For each configuration (a particular speaker architecture and $\lambda_I$), we trained 5 teams from scratch for 5,000 episodes and plotted the median and quartile rewards, smoothed over a 50 episode window.

A consistent trend emerged: higher $\lambda_I$ led to faster convergence to a higher reward. Indeed, this corroborates findings by Lin et al. [6], who used an autoencoder architecture for communication in multiagent settings, but here we showed good convergence properties when mixing utility and informativeness terms (whereas they reported worse results when adding utility losses). Overall, these findings motivated training agents to convergence with high $\lambda_I$ before beginning to penalize complexity with $\lambda_C$.

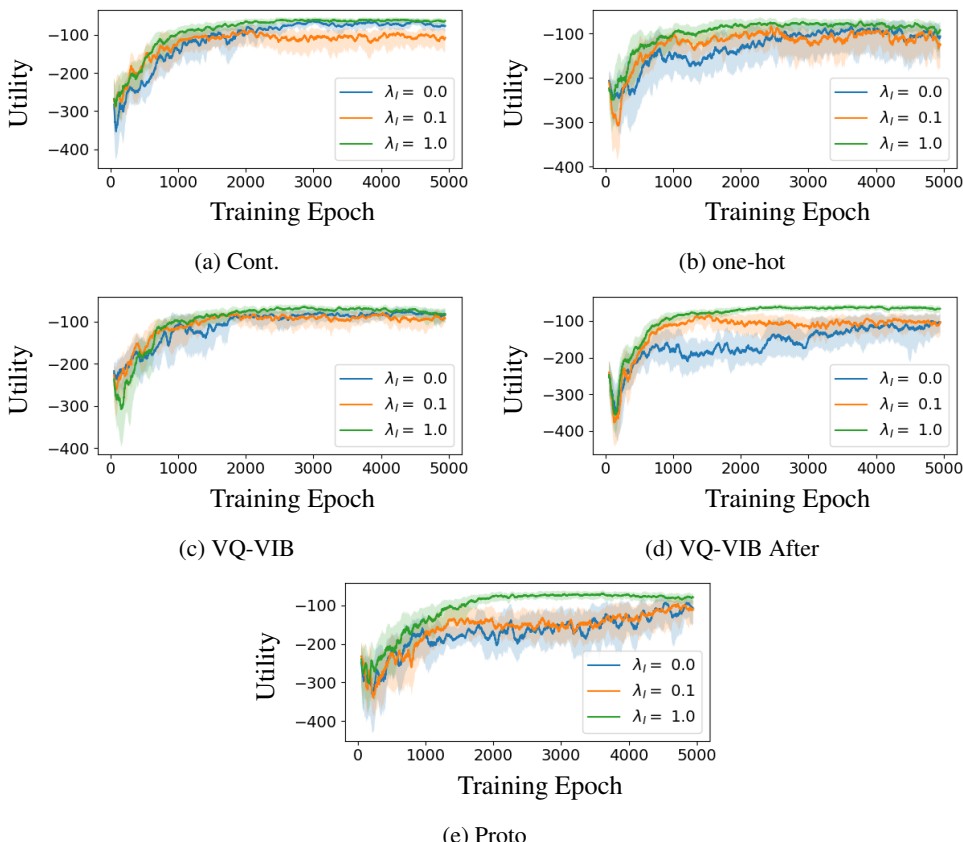

Figure E.1: In the 9 points environment, using a higher weight for informativeness ($\lambda_I$) resulted in faster convergence to higher rewards for each communication architecture. Medians and quartiles plotted over 5 trials for each configuration.

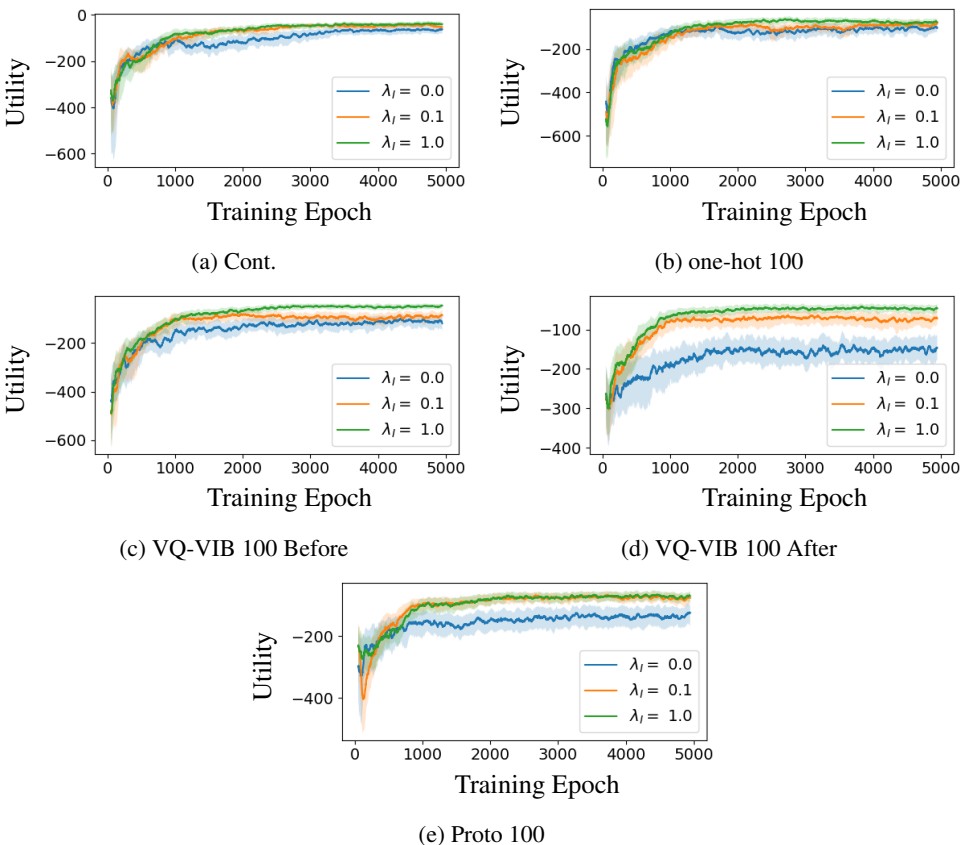

(a) Cont.

(b) one-hot 100

(c) VQ-VIB 100 Before

(d) VQ-VIB 100 After

(e) Proto 100

Figure E.2: In the Uniform environment, we observed similar trends as in 9 points: greater $\lambda_I$ improved convergence rates and rewards. Medians and quartiles plotted over 5 runs.

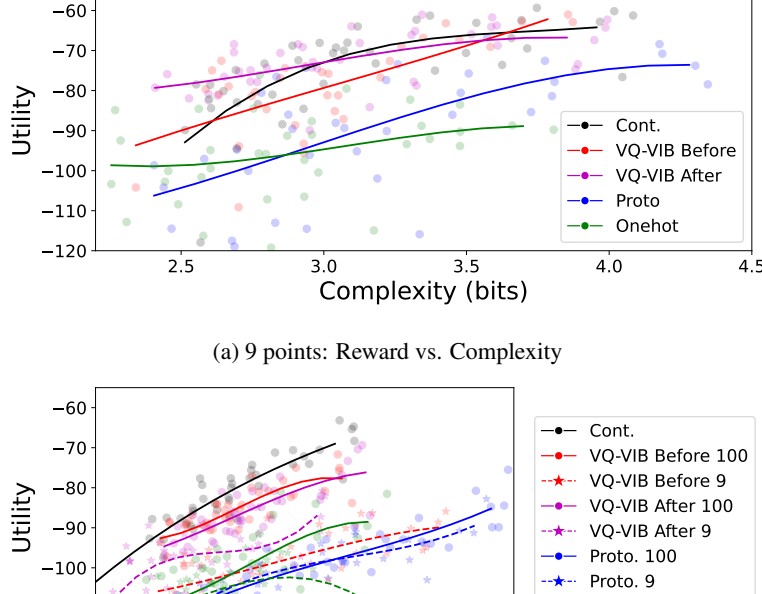

(a) 9 points: Reward vs. Complexity

(b) Uniform: Reward vs. Complexity

Figure E.3: In the 9-points and Uniform environments, agent reward (utility) decreased with complexity. Each point represents the mean complexity and reward for agent checkpoints at 1,000 episode increments during training, over 5 random seeds. Curves are the best cubic fits to the data. VQ-VIB architectures typically outperformed other discrete methods by achieving higher reward for the same complexity, and over-parametrizing with additional embeddings (100 vs. 9) also helped.

Next, we include the results of annealing complexity in both the 9 points and Uniform environments, as depicted in Figure E.3. In both environments, as we increased $\lambda_C$, complexity and utility decreased. Comparing both VQ-VIB architectures to other discrete communication methods showed that VQ-VIB agents tended to obtain a higher mean reward, for a given complexity, than other methods. In the Uniform environment, we also compared different variants of VQ-VIB, Proto, and one-hot agents based on the maximum number of discrete messages allowed by the neural architectures. We tested agents parametrized with either 100 or 9 unique messages; we hypothesized that initializing agents with many discrete messages and then gradually decreasing complexity would result in better performance than training with few messages from the start. Indeed, we observed that was the case. As seen in Figure E.3 b, VQ-VIB agents parametrized with 100 embeddings consistently achieved higher reward for the same complexity than VQ-VIB agents parametrized with only 9 embeddings. This trend held even when the complexity was lower than $\log_2(9)$ bits.

Ultimately, results from both particle world environments corroborate the findings from the color reference games: encouraging informativeness improved convergence, VQ-VIB achieved greater utility than one-hot, and decreasing complexity led to coarser discretizations (main paper, Figure 4). The varied domains and training mechanisms (MARL vs. backpropagation for reference games) leading to consistent results suggest broad generalizability of our framework and VQ-VIB.

# F    Color Reference Game Full Results

We omitted some of the color reference game results from the main paper due to space constraints. Here, we include our complete results.

## F.1    Basic Color Reference Game

First, we include results from the Basic setting, in which we trained agent with different values of $\lambda_I$ without varying $\lambda_C$. These results are included in Figure F.1. Setting $\lambda_I = 0$ induced similar conditions to those used by Chaabouni et al. [7]; any positive $\lambda_I$ favored more informative (and therefore more complex) communication. Given the monotonically increasing nature of the IB bound, an optimal agent trained with any positive $\lambda_I$ would maximize utility and informativeness by maximizing complexity, up to a maximum of $\log_2(330)$ bits. Instead, we found that agents appeared to converge to lower complexity values, although increasing $\lambda_I$ tended to somewhat increase complexity and informativeness. For example, one-hot communication increased in complexity from roughly 2.3 bits for $\lambda_I = 0.0$ to 3.0 bits for $\lambda_I = 1.0$. This alone is a novel result compared to Chaabouni et al. [7], who trained agents with no informativeness loss and observed lower-complexity communication. Curiously, VQ-VIB agents automatically converged to informativeness around 3.5 bits regardless of the $\lambda_I$ value, although increasing $\lambda_I$ did improve training stability. This difference between one-hot and VQ-VIB indicates that the differing architectures themselves have different inductive biases.

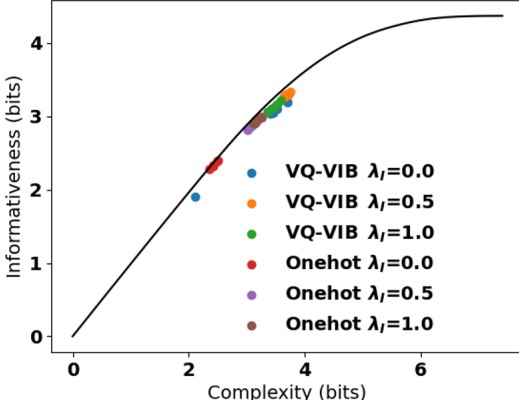

Figure F.1: Training agents in the Basic color reference game for various $\lambda_I$ but no complexity loss. Without positive $\lambda_I$, one-hot learns low-informativeness communication, but increasing $\lambda_I$ generated more complex agents, especially for one-hot.

The results in Figure F.1 were generated for varying $\lambda_I$ and low (but fixed) $\lambda_C$. We wished to train agents with no penalty for complexity, but in practice we found that one-hot (but not VQ-VIB) agents often failed to converge to any meaningful communication if $\lambda_C = 0$ throughout training. Therefore, for both VQ-VIB and one-hot agents, we initially set $\lambda_C = 0.05$ for the first 30 epochs before setting $\lambda_C = 0$ for the remaining of training. We encountered similar convergence issues when training one-hot agents to maximize utility for Table 1 (to recreate the "VQ-VIB (max $U$)" column). One-hot agents did not converge for a constant $\lambda_C = 0$ but did if trained for 30 epochs with $\lambda_C = 0.05$ before being reset to 0. We leave exploration of these initialization weights as an area for future work.

We conducted further experiments to compare agents with the neural agents presented by Chaabouni et al. [7], wherein one-hot agents were parametrized with a communication dimension of 1024. We therefore tested VQ-VIB and one-hot agents with 1024 tokens, instead of the 330 used for the results in our main paper. This hyperparameter change largely did not alter results, which is unsurprising given that agents with 330 tokens did not use their full vocabulary. As shown in Figure F.2, both VQ-VIB and one-hot achieved nearly optimal informativeness for a given complexity, and VQ-VIB achieved higher utility than one-hot (results zoomed in to complexity values between 1 and 2 bits for clearest plotting of these trends).

We performed additional experiments to investigate the effect of training with REINFORCE, instead of the supervised learning framework used in our main experiments (e.g., backpropagating through

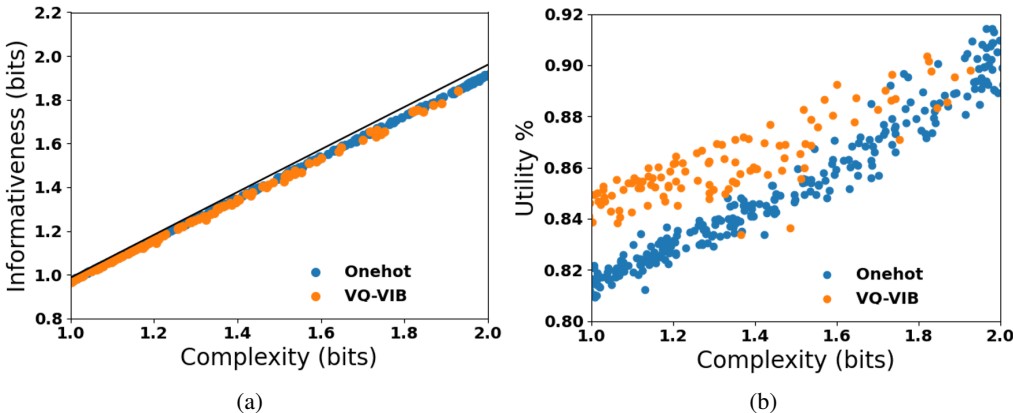

(a)                                                                (b)

Figure F.2: Communicative informativeness (a) and utility (b) as a function of complexity for color agents trained with 1024 tokens in the Basic game. These results largely reproduced results from our main paper with 330 tokens, wherein VQ-VIB and one-hot were similarly efficient, but VQ-VIB achieved higher utility than one-hot.

gumbel softmax) [8]. Chaabouni et al. [7] had found that training with REINFORCE typically resulted in lower-complexity communication than when training with messages generated via a gumbel softmax layer, and that training with REINFORCE was less stable. Results from our REINFORCE experiments, using 330 tokens for one-hot and VQ-VIB, are included in Figure F.3. We trained agents with $\lambda_U = 1.0$, $\lambda_I = 50.0$, and $\lambda_C$ was annealed from 0 to 0.1, using code adapted from Rita et al. [9].

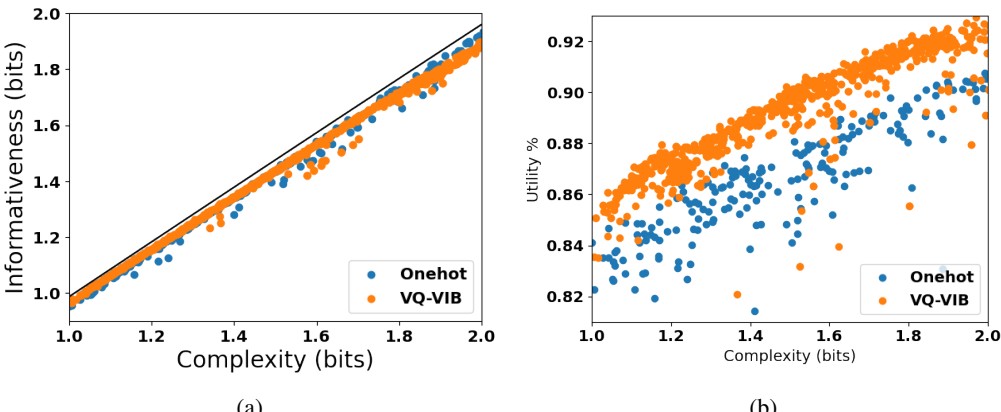

(a)                                                                (b)

Figure F.3: Informativeness (a) and utility (b) versus complexity for agents trained with 330 tokens, using REINFORCE, in the Basic reference game. These results closely tracked prior experiments: both one-hot and VQ-VIB were efficient, and VQ-VIB achieved greater utility, for the same complexity, than one-hot.

Our results from these experiments align with all prior findings: VQ-VIB and one-hot achieved high communicative efficiency, and VQ-VIB produced greater utility than one-hot did. Beyond those corroborating results, our experiments aligned with Chaabouni et al. [7]'s findings and further illustrated the importance of informativeness losses and the VQ-VIB architecture. With zero informativeness loss (i.e., $\lambda_I = 0$) one-hot agents often failed to converge to any useful communication (as indicated in prior art), but by increasing $\lambda_I$, agents consistently converged to useful communication. Indeed, with $\lambda_I = 50$ and $\lambda_C = 0$, one-hot agents learned to communication with complexity of up to 2.4 bits, greater than any complexity value found by Chaabouni et al. [7] using REINFORCE over 180 seeds (see Fig. 5 in that paper). Furthermore, we observed that VQ-VIB agents typically converged to higher informativeness (and higher complexity) communication than one-hot when complexity

was not penalized, indicating that further investigation of inductive biases is a promising direction for future work. Overall, this experiment largely replicates our findings from the main paper.

Thus, findings from our REINFORCE experiments indicate that the main conclusions of our paper hold true across several training mechanisms, and furthermore that our framework (including in particular the informativeness loss) can address issues already noted in prior work.

## F.2  Common Ground Color Reference Game

Here, we include the full results from our Common Ground color reference game experiments. Crucially, these experiments tested agent behavior when utility and informativeness were not identical: utility could be maximized by the speaker only communicating about the target color, whereas informativeness was maximized by communicating about both target and distractor. Given the lack of human-labeled data for such a setting, we were unable to compare agent communication with human languages. However, by analyzing communication when varying the target for a fixed distractor, or by varying the distractor for a fixed target, we established that agents conditioned communication both on the target and distractor colors.

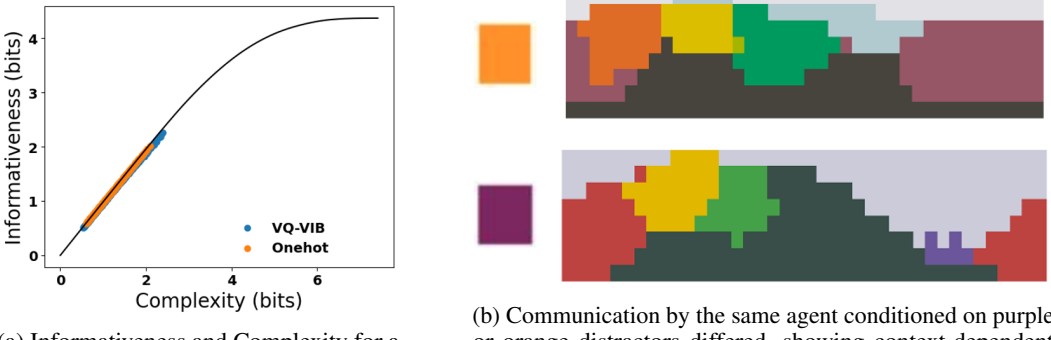

(a) Informativeness and Complexity for a gray distractor

(b) Communication by the same agent conditioned on purple or orange distractors differed, showing context-dependent communication.

Figure F.4: In Common Ground experiments, both VQ-VIB and one-hot communication demonstrated high informativeness for a fixed distractor (left), and varied communication according to distractor (right).

We trained agents in the Common Ground setting by randomly sampling target and distractor colors, as in the Basic setting, but here the speaker observed noisy versions of the distractor as well as the target. Based on insights that training to increase informativeness produced more complex communication, we trained agents for 30 episodes with $\lambda_I = 1.0$. For the next 70 episodes, we annealed the complexity weight, $\lambda_C$, using the same rates as in the Basic setting; this produced communication at different levels of complexity. As in the Basic setting, we found that VQ-VIB achieved higher utility, for the same complexity, than one-hot (Figure F.5).

We further analyzed communication in two ways: using a fixed distractor with varying targets and using a fixed target with varying distractors. First, for a fixed, gray distractor (#46 in the WCS data), we measured the standard metrics of informativeness and complexity. That is, we set the distractor color to gray, and we recorded, for each of the other 329 colors as the target, the speaker's communication. We then performed the same complexity and informativeness analysis as in the Basic setting, and found that both VQ-VIB and one-hot communication remained near-optimally efficient, as shown in Figure F.4a.

Second, we measured whether agents conditioned their communication on the distractor color. Unlike the prior evaluation, in which we varied the target color for a fixed distractor, here, we evaluated communication for different distractors. Plots of a VQ-VIB agent's naming scheme, using either an orange (color #212) or purple (color #325) distractor are shown in Figure F.4b.

As expected, the agent displayed different resolutions in regions of color depending upon the distractor. Specifically, for an orange distractor, the speaker had finer resolution around shades of orange and yellow; conversely, the speaker had finer resolution around shades of purples for a purple distractor.

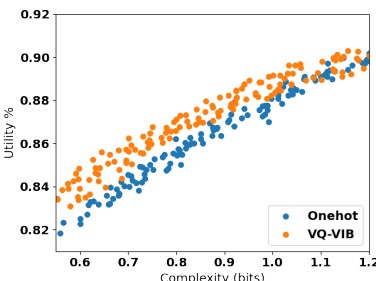

Figure F.5: In the Common Ground color reference game (as well as the Basic reference game, presented in the main paper), VQ-VIB achieved greater utility, for the same complexity, than one-hot communication.

This aligns with expectations that, when trying to differentiate between colors, one must use more complex or accurate words to distinguish similar colors.

Jointly, our findings indicate that many of the themes identified in the Basic setting – nearly-optimal communicative efficiency, the importance of annealing $\lambda_C$ to generate varied complexity, and the advantages of VQ-VIB over one-hot – held in the Common Ground setting as well.