# OpenReview forum: "Trading off Utility, Informativeness, and Complexity in Emergent Communication"
_NeurIPS.cc/2022/Conference — NeurIPS 2022 Accept_

### Official Review · Reviewer_1FYj · 2022-07-05

**Rating:** 7
**Confidence:** 4
**Soundness:** 4 excellent
**Presentation:** 4 excellent
**Contribution:** 3 good

**Summary:**

In this work, the authors introduced a new discretization method for endowing neural agents with a communication channel in a cooperative multi-agent framework. By adding auxiliary losses, each penalizing the utility, informativeness, and complexity of the generated language, testing on two MARL benchmark and a referential task, the authors study the conditions that led the communication protocol to be successful and to resemble properties of natural language as described by the IB (information bottleneck) principle.

**Questions:**

Lin et a. 2021 reports how difficult it can be to train fully decentralized agents for tasks that involve learning a communication protocol. Sharing gradients between agents is a form of communication that could not used in a fully decentralized setups. Have you considered or tried training your agents with some form of RL algorithms that would not involve any gradient passing between agents? Note that this is not to be intended as a missing experiment for this work but rather as something to be intended as future work.

**Limitations:**

The authors have adequately addressed limitations and negative societal impact of their work

**Strengths And Weaknesses:**

I believe the main strengths of this paper are in the introduction of VQ-IB and the analysis of the contribution of the additional losses in terms of complexity and informativeness.  The experimental setup is solid and the results support the idea that the introduced terms can steer the emergent languages to different complexity classes and make them similar to natural languages.

I could not find any major weaknesses in this work. Useful addition to the paper would be a qualitative analysis for the navigation environments. The quantitative results are informative and support the main claim of the paper but a qualitative analysis might help shed a light on better understanding the language protocol. Interesting questions would be of this sort: "are there specific messages the systematically mapped to actions of the environment e.g. go right, the target is in the top right corner, go there etc.".

While I believe that the tested environments correctly corroborates the introduced hypothesis, it would be interesting to know whether the discretization method can scale to larger scale setup involving complex input like natural images. Havrylov and Titov 2017 and Dessi et al. 2021 showed that agents can learn a protocol describing natural images when trained from pixel input and without any supervised pretraining. VQ-IB is a drop in replacement in their methods (they both used gumbel softmax sampling) so it would interesting to see wether your results hold at that scale. Given that the way "complexity" is penalized in previous work is usually by changing vocabulary sizes it could be useful to test whether an auxiliary loss term would be more effective instead.


Finally, I consider the introduced method and the presented results interesting, the analysis to be thorough and informative and believe this paper should be included in NeurIPS since it could be beneficial for the emergent communication community.

-----

Minor:

- A brief description of the Proto architecture would be useful to understand its differences with the other architectures

---

> ### Author Response · Authors · 2022-08-02
> **Minor clarifications for Reviewer 1FYj**
>
> We are happy that the reviewer found our contributions beneficial for NeurIPS in general and for the emergent communication community in particular.
>
> As for the reviewer’s specific comments:
>
> 1. The reviewer suggests a qualitative analysis for the navigation environments as a useful addition to the paper. We believe that this is addressed, at least in part, in Appendix C, Fig 12 (Fig 10 in the original submission), which offers a qualitative view of the results. It can be seen that the agents learn to use the signals in a way that partitions the space in a meaningful way. Due to space limitations and the breadth of this work, we were unable to include this qualitative analysis in the main text, but we are pointing the reader to Appendix C for a full description of the results.
>
> 2. The reviewer suggests extending this work by testing VQ-VIB in larger-scale settings. We absolutely agree that this is an important and exciting direction for future research, and we have actually started to explore this as a follow-up project. Results in this paper suggest that VQ-VIB might be better-suited to high-complexity domains than onehot. For example, in the color reference game, VQ-VIB learned high-informativeness, high-complexity communication even without an explicit informativeness loss, whereas onehot tended to lower-informativeness communication. Thus, VQ-VIB might be most useful compared to prior art in even more complex domains; we have highlighted the importance of such future work in Section 6.
>
> 3. We have included a short summary of the Proto architecture in Section 4.2, in addition to the architecture details already provided in Appendix A.1.
>
> 4. The reviewer’s suggestion to consider more challenging RL settings without gradient-passing is important. We are already exploring such methods via a REINFORCE-based implementation of the color reference game experiments, as suggested by reviewer KUCM. However, given convergence challenges associated with REINFORCE, we do not yet have results ready for the rebuttal; we will include them in the camera-ready version if this paper is accepted.

---

> > ### Comment · Reviewer_1FYj · 2022-08-03
> > **Response**
> >
> > Thanks for your reply. Here are some comments:
> >
> > 1 - I find that figure interesting but not a deep qualitative analysis of the agents protocol. If anything, I think it corroborates the complexity result already presented, I don't think it properly addresses questions like the one I mentioned: "are there specific messages the systematically mapped to actions of the environment e.g. go right, the target is in the top right corner, go there etc."
> >
> > 4 - Chaabouni et al had experiments with Reinforce, if you are already running experiments with, at least with the onehot setup it should be feasible to report observed performance. This would at least confirm the transfer of your new losses to a different optimization scheme.

---

> > > ### Author Response · Authors · 2022-08-04
> > > **REINFORCE experiment results**
> > >
> > > We thank the reviewer for the clarification on qualitative analysis; we will try to develop such tools in the coming days.
> > >
> > > We have updated the paper to reflect findings from our REINFORCE experiments, which have now concluded. Findings are included in Appendix D and are most visible in Figure 15. We largely reproduced findings from earlier in our paper. In particular, comparing VQ-VIB and onehot showed similar communicative efficiency, but VQ-VIB achieved greater utility. We also found that training with an informativeness loss overcame many of the issues identified by Chaabouni et al.. That is, with $\lambda_I = 50$, agents consistently learned non-degenerate communication, whereas Chaabouni et al. used 180 random seeds to overcome training failures. Thus, we believe that these results provide still further evidence for the advantages of our framework and neural method.

---

> > > > ### Comment · Reviewer_1FYj · 2022-08-08
> > > > **Response to new experiments**
> > > >
> > > > I thank the authors for running additional experiments and performing and including the new protocol analysis.
> > > >
> > > > I understand some of the other reviewers' concerns and especially think the novelty might be limited for people outside the context of the emergent community, but, nonetheless, I confirm the score that I assigned it in my initial review.
> > > >
> > > > ---
> > > > Minor:
> > > > line 726: Chaabouni et al. [35] had found that training with REINFORCE typically resulted in lower-complexity communication than with onehot, and that training was less stable. --> I understand the definition onehot makes sense within the context of gumbel-smax experiments but I found it confusing when comparing with REINFORCE. I would maybe say something along the lines of "lower-complexity communication than with gumbel-softmax training with a fully connected layer generating message representations?

---

> > > > > ### Author Response · Authors · 2022-08-09
> > > > > **Minor rewording**
> > > > >
> > > > > Thank you so much for confirming your initial score arguing for acceptance!
> > > > >
> > > > > As for your final minor comments, we have changed the phrasing in our latest revision to clarify this sentence. It now states that Chaabouni et al. “... found that training with REINFORCE typically resulted in lower complexity communication than when training with messages generated via a gumbel softmax layer…”

---

> > > ### Author Response · Authors · 2022-08-05
> > > **Qualitative Analysis Followup**
> > >
> > > Based upon the reviewer's suggestions, we have included new qualitative analysis of particle-world communication in the revised paper (page 20, Figure 12). Our new analysis shows how tokens referred to regions in the Uniform environment 2D space, and that lower-complexity communication created larger regions, or coarser discretizations. Such analysis also highlights deep parallels with the color reference game, in which agents discretized a continuous (color space). The visualizations we now include make this connection more obvious; we thank the reviewer for this insightful suggestion.

---

### Official Review · Reviewer_3YrT · 2022-07-13

**Rating:** 4
**Confidence:** 4
**Soundness:** 3 good
**Presentation:** 4 excellent
**Contribution:** 1 poor

**Summary:**

This paper analyzes the influence of three factors on languages emerging in communication games between simple VQ-VAE based agents. Specifically, the authors contrast the effect of promoting utility and "informativeness" versus reducing the complexity of the language. The authors demonstrate that informativeness generally improves success at the game, but at the detriment of complexity. An adequate tradeoff between these two factors seem to lead to "human-like" languages, at least as measured by a comparison to color-naming schemes.

**Questions:**

- What is semantic communication? Or rather what would the opposite of semantic communication be? Communication without meaning?
- What is the purpose of VQ-VIB after? It seems to me that randomness in VQ-VIB after emulates a perturbation in the environment (on which the agent has no control), whereas randomness in VQ-VIB before models uncertainty from the agent (via a reparametrization trick). It is not clear to me why the former is interesting in the context of this work (in fact it is used much less than VQ-VIB before in the experiments.
- What are $m$ and $\hat m$ in section 3.1? Are they typos for $x$ and $\hat x$?

**Limitations:**

I think there could be at the very least a more thorough discussion of the notion of complexity of a language (and the difficulty of defining a metric)

**Strengths And Weaknesses:**

While the paper touches on interesting issues (the interplay between informativeness and complexity), I am having a hard time understanding what I should make out of it. The extreme simplicity of the tasks and languages considered, coupled with a debatable definition of complexity make it hard to understand the relevance of the results.


### Strengths

1. **Clear exposition**: the paper is rather clear throughout. The writing reads well and a number of well designed figures illustrate the author's points (in particular figure 1)
2. **Well laid out experimental section**. I enjoyed the structure of the experiments section, first spelling out hypotheses, experimental settings and then discussing results.

### Weaknesses
1. **Notion of complexity**: In the paper, the complexity of a language is defined as the mutual information between "inputs" (what is communicated about) and signals (what is communicated). I am not entirely convinced by this notion of complexity for several reasons
    - It seems to me that a language can have a high mutual information by virtue of being unambiguous: indeed if we write $I(X;C) = H(X) - H(X\mid C)$, then for a fixed distribution over inputs, the mutual information is maximal for languages where $H(X\mid C)=0$, in other words when any given message $c$ refers unambiguously to a single input $x$. To me this seems unrelated to complexity. For example emergent languages might be equally unambiguous, yet more "complex" in a different sense (eg. less compositional, using longer words...).
    - More generally this is an odd definition in the context of human languages. Much ink has been spilled in linguistics on the topic of language complexity: syntactical complexity vs morphological complexity, are all human languages equally complex, is there a meaningful measure of language complexity, etc... There is very little mention of this literature which is a bit of a shame given that one of the objectives of the paper is to "[demonstrate] how fundamental principles that are believed to characterize human language evolution may inform emergent communication in artificial agents." I recommend "On the Feasibility of Complexity Metrics" by Miestamo (2004) as a starting point for a discussion. It should also be mentioned that previous work in emergent communication has looked at other metrics which are related to complexity: length of the messages ("“LazImpa”: Lazy and Impatient neural agents learn to communicate efficiently", Rita et al. 2020), ease of cultural transmission ("Emergence of Compositional Language with Deep Generational Transmission", Cogwell et al. 2019, "Compositional languages emerge in a neural iterated learning model", Ren et al. 2020).

    Overall it seems to me that the notion of complexity advocated by the author is more or less the entropy of the distribution of the messages, which I understand to be a proxy for the size of the lexicon (the number of utterances in the language). I am not sure if this is an interesting metric: arguably a key feature of human languages is that they allow for generating a potentially infinite number of utterances, and what differentiate them is how they are able to do so by the use of finite means (phonemes, words...)
2. **Unclear takeaway**: What do the results of the paper mean? The two takeaways I read from the papers are that (1) "principles that characterize language evolution can inform emergent communication" and (2) "penalizing complexity is necessary to avoid idiosyncratic languages". I would contest that (1) is a novel insight because there is already plenty of work porting ideas from cognitive science and language evolution to emergent communication (see work on iterated learning, zipf's law in emergent languages, etc...). As for (2) I am not entirely convinced due to my issues with the definition of complexity used in the paper (outlined above).
3. **Informativeness vs utility**: The distinction between informativeness and utility does not seem very useful in the context of the paper: indeed the two are not really contrasted in experiments. In a lewis signaling game (such as the color naming experiment) they are essentially equivalent. It would have been interesting to also vary along this dimension in experiments. Alternatively, I think the key claims of the papers could have been made by focusing on the opposition between informativeness and complexity (and leaving utility out of the picture).

---

> ### Author Response · Authors · 2022-08-02
> **Response Part 1: Clarifications on Complexity and Takeaways**
>
> We begin by addressing the points that are considered by the reviewer as weaknesses, and then we respond to the reviewer’s specific questions.
>
> ### 1. Notions of complexity:
>
> First, we would like to clarify the motivation for our complexity term. From a theoretical perspective, the use of mutual information for measuring complexity is grounded in information theory (e.g., [Shannon 1959](https://www.gwern.net/docs/cs/algorithm/1959-shannon.pdf); [Tishby et al., 1999](https://www.cs.huji.ac.il/labs/learning/Papers/allerton.pdf); [Gilad-Bachrach et al. 2003](https://link.springer.com/chapter/10.1007/978-3-540-45167-9_43)) and has been widely studied in machine learning (e.g. [Bialek et al., 2001](https://www.princeton.edu/~wbialek/our_papers/bnt_01a.pdf)), neuroscience (e.g. [Tkacik & Bialek, 2016](https://www.annualreviews.org/doi/abs/10.1146/annurev-conmatphys-031214-014803)), and cognitive science (e.g. [Sims 2018](https://www.science.org/doi/abs/10.1126/science.aaq1118)). In the context of linguistics, this definition has recently been shown to capture the **semantic complexity of the lexicon**, i.e., the complexity of how words partition the environment into categories. This idea was introduced by Zaslavsky et al. (PNAS, 2018), and has been gaining empirical support across hundreds of languages and multiple domains  ([Zaslavsky et al. 2018](https://www.pnas.org/doi/full/10.1073/pnas.1800521115), [2019](https://arxiv.org/pdf/1905.04562.pdf), [2021](https://escholarship.org/uc/item/2sj4t8m3),  [2022](https://academic.oup.com/jole/advance-article-abstract/doi/10.1093/jole/lzac001/6566271?redirectedFrom=fulltext&login=false); [Mollica et al. 2021](https://www.pnas.org/doi/full/10.1073/pnas.2025993118)).
>
> We agree that there are various notions of complexity in linguistics and that it is not always clear how to define complexity. While this is an important question, it is not the one this paper aims to tackle. The goal of this paper is to test whether the framework of Zaslavsky et al. for lexical semantics, which predicts a specific complexity measure, can inform emergent communication (EC) in AI. Having said that, we have extended Section 2 to acknowledge the debate about complexity and included the references suggested by the reviewer.
>
> **Intuition for our complexity measure**:
>
> 1.1. Complexity and ambiguity: indeed, unambiguous language are maximally complex. However, in order to achieve this in our setup, agents must assign a unique signal to every possible referent, which may require a huge lexicon. Thus, if agents have limited resources (as humans do), they must compress inputs into signals in order to reduce the complexity of their lexicon. Our complexity term measures the degree of compression (Tishby et al., 1999). In our setting, only one signal can be transmitted in each round and all signals have the same length, thus other notions of complexity that are related to compositionality or word-forms are not applicable in this case. Note that we focus primarily on **lexical semantics** rather than on other aspects of language such as syntax or morphology.
>
> 1.2. Complexity vs. entropy: while our complexity term is related to entropy and lexicon size, it is also fundamentally different. For example, Zaslavsky et al. (2018) showed that using lexicon size as a complexity measure yields qualitatively wrong predictions for human color naming, in contrast to the use of mutual information which currently gives the SOTA model in this context. Entropy only considers the marginal distributions of signals, which ignores the fact that languages often use non-deterministic signaling patterns.
>
> ### 2. Unclear takeaways:
>
>  As summarized in Section 6, our key contributions are: (a) finding that combining informativeness and complexity in training EC led to faster convergence and human-like communication systems; and (b) our new neural architecture, VQ-VIB, outperformed existing discrete communication methods.
>
> We are certainly not the first to advocate for applying notions from cognitive science to EC, but we are (to the best of our knowledge) the first to integrate utility, informativeness, and complexity during training. This approach is motivated by the recent body of literature on the Information Bottleneck (IB) framework for semantic systems (Zaslavsky et al. 2018), which has not previously been incorporated directly into training EC, as we do here.

---

> > ### Author Response · Authors · 2022-08-02
> > **Response Part 2: Clarification on Informativeness vs. Utility, and Specific Questions**
> >
> > ### 3. Informativeness vs. utility:
> >
> > While informativeness and utility are indeed related, they differ in important ways. Generally, utility focuses on a specific task, whereas informativeness is task-agnostic. For example, Chaabouni et al., in their reference game experiments, only trained agents with a utility loss, so they could only affect communication by varying the environment. Conversely, our approach allows leaving the training environment (and therefore utility pressures) unchanged but varying informativeness and complexity to get a wider range of communication systems. Our results show the benefits of this approach, and in particular, the importance of considering informativeness in addition to utility alone. Specifically, we found that training with informativeness, in addition to utility, can yield higher overall utility and faster convergence (Figs 4a, 9, and 10), and that controlling complexity was necessary for learning communication schemes that reflected the varied complexities in the world’s languages (Fig 5, Table 1). In the Common Ground experiments, we explicitly tested agents in settings where utility and informativeness were not aligned. Informativeness would be maximized by the speaker communicating about both colors; utility could be maximized by the speaker communicating about only the target. Nevertheless, training with informativeness improved agent utility yet again in this setting. We have emphasized these differences in our revised paper, in Sections 5.2.1 and 5.2.2.)
> >
> > The reviewer suggests that it “would have been interesting to also vary along this dimension” (informativeness) or focus on informativeness vs. complexity. Indeed, our results already examine the tradeoffs between these dimensions by varying either $\lambda_I$ or $\lambda_C$ (Figs 4b, 5, 9, 10, 11a, 13, and 15a).
> >
> >
> >
> >
> > ### **Specific questions**
> >
> > i. What is “non-semantic communication?” Research in other areas of linguistics like syntax and phonology consider aspects of communication that are related to form rather than meaning. We focus on semantic communication in this work, i.e., the way signals partition the environment into semantic categories.
> >
> > ii. Reviewer kucm also questioned the point of VQ-VIB After. We appreciate this feedback and have moved this formulation from the main paper to an appendix in the revised submission. We initially developed VQ-VIB After as a baseline for comparison, and we agree that it is less interesting than VQ-VIB Before.
> >
> > iii. “What are $m$ and $\hat{m}$?” These variables correspond to probability distributions (belief states) over the feature space, as indicated on line 110. This notation borrows from Zaslavsky et al.’s variables for “meanings.” Intuitively, we think of there being some true state, $x$, and an agent’s belief over that state, $m$, which may be shaped by perceptual noise.

---

> > > ### Comment · Reviewer_3YrT · 2022-08-08
> > > **Response to Rebuttal**
> > >
> > > I thank the authors for their response. Overall let me restate that I think the paper touches on interesting issues, but some of my concerns remain with respect to its situation in the emergent communication literature.
> > >
> > > > Complexity
> > >
> > > I appreciate that the authors added a more extensive discussion of the concept of complexity in the related work. I do think that the emphasis on "lexical communication systems"$^1$ and the associated information-theoretical notion of complexity could be made clearer earlier on in the paper (abstract/introduction). As it stands I think the opening is a bit too broad, and the connection to the current emergent communication literature is not very clear.
> > >
> > > An example: in the introduction it is stated that "[emergent communication] may lead to communication systems that are too complex for humans to understand [7]" with a reference to Chaabouni et al. 2019 "Anti-efficient encoding in emergent communication". However, that paper does not really consider complexity as studied in this submission: they look at the distribution of message lengths rather than the size of the vocabulary. Furthermore, they specifically only study settings where the channel capacity (~ $\text{vocab. size}^\text{message length}$) is large enough for every input x to be assigned a single message. This is problematic since this reference to Chaabouni et al. 2019 is used to motivate the rest of the paper.
> > >
> > > More generally I can think of a number of works studying idiosyncratic properties of emergent communication systems (e.g. non-compositionality [1], perceptual shortcuts [2]), but it is not clear to me how the problem they study align with the problem studied in this paper. Perhaps they don't, and this is fine, but in this case the authors should make a stronger point to motivate why addressing the issue of complexity in color-naming systems has any bearing on emergent communication (or machine learning, or computational linguistics, but the current framing of the paper is very much focused on emergent communication).
> > >
> > > Overall I think more work is needed in the introduction to properly situate this paper's contribution in the context of current issues in the emergent communication literature.
> > >
> > > $^1$I would be a bit wary of using the term "lexical semantics" as it generally refers to the study of the meaning of individual words in natural languages)
> > >
> > > ## Further comments
> > >
> > > > VQ-VIB after moved to appendix
> > >
> > > Thanks, I think the paper is much clearer this way.
> > >
> > > > The reviewer suggests that it “would have been interesting to also vary along this dimension” (informativeness) or focus on informativeness vs. complexity. Indeed, our results already examine the tradeoffs between these dimensions by varying either  or  (Figs 4b, 5, 9, 10, 11a, 13, and 15a).
> > >
> > > I meant that it would have been interesting to study the informativeness-utility trade-off further. I am aware that the paper performs multiple experiments on informativeness vs. complexity: my point was that this could have been the core focus of the paper (ie the title could have been "Trading off Informativeness and Complexity in Emergent Communication"). This is a relatively minor point though
> > >
> > > > $\hat m$ / $m$ vs $x$, $\hat x$
> > >
> > > Thanks for clarifying, I had missed this in my first reading. That being said, is this distinction really useful in the context of the paper? $m$ and $\hat m$ are not paper later on and their introduction might be more confusing than anything else.
> > >
> > > If the distinction is critical, it might be worth reusing the notation elsewhere in the paper where appropriate (perhaps in figure 1?). Otherwise I think this notation can be removed
> > >
> > > > Semantic communication
> > >
> > > To clarify my point: I think "semantic communication" is a pleonasm. You are right that other areas of linguistics are concerned with other aspects of **language** such as syntax, morphology, discourse, etc... But syntax for example is not integral to communication: for example the communication system studied in this paper is not syntactic. In this regard it makes sense to refer to "syntaxic communication" to denote systems of communication that exhibit syntax (see e.g. [3]). A similar reasoning would hold for morphology.
> > >
> > > However I am hard pressed to find examples of communication systems that do not exhibit semantics, i.e. communication systems that do not convey meaning. As far as I know, "semantic communication" is not a standard term in linguistics (it is certainly not used by Zaslavsky and colleagues).
> > >
> > > How about just using "communication"? Or perhaps if the emphasis is to be on semantic, why not use "semantic categorization (systems)"?

---

> > > > ### Comment · Reviewer_3YrT · 2022-08-08
> > > > **Response to rebuttal (references)**
> > > >
> > > > ## References:
> > > >
> > > > - [1]: Chaabouni, Rahma, et al. "Compositionality and Generalization In Emergent Languages." Proceedings of the 58th Annual Meeting of the Association for Computational Linguistics. 2020.
> > > > - [2]: Bouchacourt, Diane, and Marco Baroni. "How agents see things: On visual representations in an emergent language game." Proceedings of the 2018 Conference on Empirical Methods in Natural Language Processing. 2018.
> > > > - [3]: Nowak, Martin A., Joshua B. Plotkin, and Vincent AA Jansen. "The evolution of syntactic communication." Nature 404.6777 (2000): 495-498.

---

> > > > > ### Author Response · Authors · 2022-08-08
> > > > > **Clarification of complexity in prior art**
> > > > >
> > > > > We thank the reviewer for their response; reading it has clarified some of our questions about their initial review, and our new revision better situates our contributions (Introduction lines 24-27, 39-40; Related Work lines 76-78, 92-96; Experiment Preliminaries line 246-249).
> > > > >
> > > > > The key change in our revision is highlighting a different Chaabouni et al. paper than the one noted by the reviewer, which perhaps led to some of our confusion in discussions. We now highlight ["Communicating artificial neural networks develop efficient color-naming systems"](https://www.pnas.org/doi/full/10.1073/pnas.2016569118) by Chaabouni et al. (PNAS 2021) instead of "Anti-efficient encoding in emergent communication" by Chaabouni et al. 2019.
> > > > >
> > > > > In "Communication artificial neural networks..." (henceforth [1]), Chaabouni et al. study the complexity and informativeness of emergent communication in a color reference game, as in our work. Thus, [1] is perhaps the best motivation and comparison point for our paper, and we have sought to compare our work more explicitly to theirs in our revision. In particular, we highlight that in our framework, we generated a greater diversity of communication complexity than [1], and we did so without having to change the training environment. Conversely, Chaabouni et al. [1] varied how distractor images were selected to induce different levels of complexity, while we feel that optimizing for complexity directly is a much more natural and principle-based method.
> > > > >
> > > > > More generally, we note that studying the complexity and informativeness of emergent communication is well-motivated and already considered important in prior literature, as evidenced by [1] and subsequent citations. We seek to fit within that area of research and thank the reviewer for helping us clarify this point.

---

> > > > > ### Author Response · Authors · 2022-08-09
> > > > > **Further clarification of terminology and situating our work**
> > > > >
> > > > > We thank the reviewer for the helpful follow-up comments. We’re happy that the reviewer found our work overall interesting, and we hope that our response below will address all of the reviewer’s concerns. Given that the main concerns appear related to situating our work within the emergent communication literature, which we were able to address in the paper with a few clarifications in the introduction and related work sections, we hope that the reviewer will consider to recommend acceptance.
> > > > >
> > > > > In our last response we addressed the reviewer’s concern about using Chaabouni et al. 2019 as a motivation for our work. That paper is indeed not central to our motivation and we have adjusted the paper to clarify that. Below we address the other concerns raised in the reviewer’s response.
> > > > >
> > > > >
> > > > > > ... but it is not clear to me how the problem they study align with the problem studied in this paper. Perhaps they don't, and this is fine, but in this case the authors should make a stronger point to motivate why addressing the issue of complexity in color-naming systems has any bearing on emergent communication (or machine learning, or computational linguistics, but the current framing of the paper is very much focused on emergent communication).
> > > > >
> > > > > We are not aware of any prior work that directly leverages the empirical evidence that human languages evolve under pressure to optimize the IB complexity-accuracy tradeoff, in order to inform EC in artificial agents. This empirical evidence is a central part of our motivation, as explained in the second paragraph of the introduction. The color naming domain is a key example in this context, which has been the focus of many studies in machine learning and computational linguistics (e.g., Steels & Belpaeme 2005; Monroe et al., 2017; Chaabouni et al. 2021; all cited in our paper). Therefore, we believe it is of interest to the NeurIPS community. To clarify, we are not “addressing the issue of complexity”, but rather proposing to view EC through the lens of a complexity-informativeness-utility tradeoff.
> > > > >
> > > > > > I would be a bit wary of using the term "lexical semantics" as it generally refers to the study of the meaning of individual words in natural languages)
> > > > >
> > > > > We have not used that term in the paper, but rather “human-like lexical communication systems”. For further clarity, we have changed that to “human-like semantic systems”, because semantic systems (short for systems of semantic categories) is a standard term that has previously been used in the literature.
> > > > >
> > > > > > Regarding the distinction between $m$ and $\hat{m}$
> > > > >
> > > > > It is important for implementation as well as for consistency with prior work on the IB framework for semantic systems.
> > > > >
> > > > > > Regarding the term “semantic communication”
> > > > >
> > > > > we have changed this term to “semantic systems”, which is a more standard term.

---

### Official Review · Reviewer_kucm · 2022-07-15

**Rating:** 5
**Confidence:** 4
**Soundness:** 3 good
**Presentation:** 3 good
**Contribution:** 3 good

**Summary:**

This work proposes to learn to communicate while balancing three different losses. The effectiveness of communication (utility as usual), the informativeness of communication as measured by autoencoding loss, and the complexity as measured by the entropy of the communication distribution. Furthermore, the authors propose a new algorithm VQ-VIB which integrates the information bottleneck with a discrete gradient estimator VQ-VAE. The authors demonstrate how to learn to communicate with their novel method and how to integrate all three losses into their method.

The authors test the three losses and their novel algorithm on two environments, a particle world where a sender must give a location to a receiver and the World Colour Survey, following Chaabouni et al, where agents must learn a vocabulary to communicate colours. In the former, the use MADDPG and compare to regular discrete, continuous, and proto communication. In the latter they compare to just discrete communication using the gumbel-softmax estimator. They find that the autoencoding loss improves performance, and the complexity loss can be used to reduce complexity. In the particle world, VQ-VIB can outpeform a regular discrete method for the same vocabulary size but not a continuous communication method. In the WCS, agents can learn an optimal language (complexity for effectiveness) as in Chaabouni et al and the complexity loss can vary across the range of complexities. In some configuration, VQ-VIB learns a language that better corresponds to the space of colours than regular gumbel-softmax.

**Questions:**

why use VQ-VIB after if it generates a continuous message and is worse than simple continuous messages?

for 5.1, you haven't demonstrated that VQ-VIB is better than one-hot, please test vocab size > 100
- if performance increases as vocab size increases, why not increase vocab size more?
- you can make the argument that VQ-VIB outperforms one-hot *at the same vocab size* but not at the sample complexity. If one-hot can reach the same performance/complexity at a larger vocab size then you can't say it completely outperforms it
- furthermore, neither agent is reaching optimal performance!
- if you can show than VQ-VIB outperforms one-hot accuracy/complexity for *any* one-hot hyperparameters, then your method becomes much more interesting and your results are stronger

related to the above, Fig 4b shows that continuous communication (cont) clearly outperforms discrete communication (one-hot) but the only difference between the two channels should be the bandwidth
- for a fairer comparison, you should increase the discrete vocabulary size (or use a channel of length > 1)

why does Fig 4b show unconverged protocols?
- ``complexity and reward measured at each increment of 1,000 training episodes from 3,000 to 10,000''
- why are you measuring unconverged protocols during training? they are obviously going to be sub-optimal?

for 5.2, please test REINFORCE for discrete communication
    - Chaabouni et al found that REINFORCE learns less complex protocols than Gumbel-Softmax which is what you use
    - REINFORCE is as common (or more common) than GS for emergent communication, so it would be the better baseline for complexity

Fig 5a is suspicious
- either 5a is too small and therefore we can't see the variance of results
- or the learned protocols are at the exact optimal accuracy / complexity tradeoff and $lambda_I = 1$ is sufficient to achieve this (despite Chaabouni et al's results which show a variance when $lambda_I = 0$)
- these results could imply the choice of a vocabulary size (or other hyperparameters) that guaranteed optimality (see Natural Language does not Emerge Naturally (Kottur et al)) and this graph should preferably me more zoomed in and likely should stick with the original vocabulary size 1024 used in Chaabouni et al


### citation suggestions

your auxiliary loss for informativeness seems the same as AEComm (Learning to Ground Communication with Autoencoders, Lin et al, 2021) and may be worth a citation
- notably, their experiments found that adding informativeness (AE) to utility (RIAL) reduces performance so your results are interesting and novel from that perspective!

please cite Concrete Distribution (Maddison et al, 2016) on top of Jang et al for the Gumbel-Softmax estimator (they are concurrent work)


### clarification
is the entropy regularization done in place of $\lambda_c I(X,C)$ or on top of that loss? (I understood it to be in place of that loss)

is the entropy regularization in one-hot equivalent to regular entropy regularization done in previous RL works (e.g. Mnih et al 2016, Lazaridou et al, 2018) or is it the opposite? Are you penalizing entropy or encouraging entropy?


in 5.2.1 you meantion that VQ-VIB learn more complex communication, did you mean the opposite since we are trying to learn less complex, more effective communication

**Strengths And Weaknesses:**


The paper is both interesting and well-written. It contains many novel insights and experiments but overall, it is not well organized and the overall message and novelty are not clear. If the authors could add some experiments listed in the questions section and perhaps formulate their contribution more clearly, I would be likely to recommend acceptance.

To begin, the paper proposes two separate ideas. The first is the three axes (or two auxiliary losses) utility, informativeness, and complexity. Utility is a given in emergent communication, and the latter two have been investigated separately before (informativeness is autoencoding in AEComm, complexity has been penalized as far back as Mordatch and Abbeel, 2017). The second idea is VQ-VIB which is a clever combination of VQ-VAE and VIB and does seem quite interesting. The issue is that these two contributions are quite separate as far as I can tell and neither contribution is very strong.

For VQ-VIB, although the algorithm is interesting it is not empirically verified in depth. The authors claim it achieves greater utility and informativeness for the same complexity but this is only true in the particle envs and not in the WCS (and could depend on hyperparamters). Furthermore, they do not do an exhaustive comparison of baselines (REINFORCE instead of GS, trying a wider range of vocabulary sizes, etc..). Where the results are better, they are not hugely better and for a novel algorithm it should be investigated on more environments or more in depth.

For the tradeoff of utility, informativeness, and complexity, the analysis is quite good and the results are interesting. Sadly, the novelty is not quite there. Demonstrating that a complexity loss can reduce complexity is not very novel and does not require so much attention (both Fig 4b and 5). Chaabouni et al's work on WCS was novel because it showed that the natural inductive bias of neural networks was similar to that of humans (low complexity, high informativeness). This work explicitly adds those factors as losses and shows it can find the optimal curve, which is again not very novel. Still, the auto-encoding loss is shown to be effective and this is quite exciting but it is not investigated in depth and no previous benchmarks are improved upon (e.g. MADDPG's particle envs)

Simply showing that these losses work as intended is not particularly interesting and as a reviewer, I am not fully convinced of the value of VQ-VIB. Because of the quantity of new ideas here, none of them seem to be investigated in enough depth either and many results are relegated to the appendix with only one or two sentence summaries. This is not very convincing and the overall paper feels like it brushes over key contributions.

Overall, a stronger contribution would be to focus on one of the two proposed ideas or demonstrate that both ideas are crucially necessary towards some goal. E.g. if the authors were working towards human-AI communication and demonstrate that both techniques (VQ-VIB, and losses) improve zero-shot human understanding of self-play messages. In that case, contributions feel like two separate but important tools towards a single goal. In the current case, there is an interesting investigation but none of the results are sufficiently convincing or novel.

---

> ### Author Response · Authors · 2022-08-02
> **Response Part 1: High-Level Clarifications on Framing and Experiments**
>
> We are happy that the reviewer is inclined to recommend acceptance given additional experiments and clarifications. We have addressed the reviewer's suggestions and concerns, as detailed below.
>
> ### I. Framing
> The reviewer frames our paper as proposing two, unconnected ideas; however, these two contributions are profoundly linked and jointly, they establish a principled framework for emergent communication (EC) and a new method (VQ-VIB) that improves upon prior art. Our key insight is to **integrate EC with the Information Bottleneck (IB) framework for the evolution of semantic systems**, which has been gaining broad empirical support in cognitive science and computational linguistics ([Zaslavsky et al. 2018](https://www.pnas.org/doi/full/10.1073/pnas.1800521115), [2019](https://arxiv.org/pdf/1905.04562.pdf), [2021](https://escholarship.org/uc/item/2sj4t8m3),  [2022](https://academic.oup.com/jole/advance-article-abstract/doi/10.1093/jole/lzac001/6566271?redirectedFrom=fulltext&login=false) , [Mollica et al. 2021](https://www.pnas.org/doi/full/10.1073/pnas.2025993118)). We do so by adopting the (task-specific) utility term from the EC literature, and deriving the complexity and informativeness terms from the IB principle of [Tishby et al. (1999)](https://www.cs.huji.ac.il/labs/learning/Papers/allerton.pdf). Within this framework, we propose a novel architecture (VQ-VIB) that supports the IB objective function. Much like VIB, which combines an objective function (a bound on the original IB objective) and an architecture that supports it, we propose VQ-VIB in order to implement VIB for discrete (rather than continuous) communication. We show that VQ-VIB outperforms previous discrete communication methods, which means that it is better suited for EC in our framework.
>
> ### II. Additional Experiments
> We are happy that the reviewer finds VQ-VIB interesting and appreciate the request for additional evaluation. We have done so in the revision (details under “specific questions,” below). While we feel that the results are stronger now, we would like to clarify that our original evaluation was also extensive: we considered several prior methods, including the popular onehot approach and the Proto approach ([Tucker et al., NeurIPS 2021](https://arxiv.org/pdf/2108.01828.pdf)), which is most relevant to our work. We also considered two variants of VQ-VIB, varied the $\lambda$ hyperparameters, and varied the vocabulary size in the particle envs. While **our work focuses on discrete communication**, we also considered a continuous comm model, which is not ecologically valid, but offers an estimate of the upper limit of communication regardless of human constraints. We are not sure why the reviewer argues that “no previous benchmarks are improved upon,” as we have shown that VQ-VIB outperforms other discrete methods (it achieves >= utility for the same complexity), both in the particle envs and in the WCS setting (Figs 6, 11, and 14b, and Table 1).
>
> We would also like to highlight that the goal of this work is to integrate EC with an **empirically-validated first-principles approach** to language evolution. This is primarily a **conceptual scientific-driven contribution**, rather than a purely engineering-driven contribution. Therefore, we find it noteworthy that our approach is able to improve on prior art, and these improvements - even if not huge - suggest that our framework has the potential of significantly advancing the field toward a better understanding of how natural languages may evolve in artificial agents.

---

> > ### Author Response · Authors · 2022-08-02
> > **Response Part 2: High-Level Clarifications on Novelty and Key Contributions**
> >
> > ### III. Novelty
> > The reviewer is concerned about the novelty of our proposed objective function and results. First, to our knowledge, this is the first study that integrates utility maximization with the IB principle in the context of EC, in contrast to treating these components separately. In addition, as far as we know,  Mordatch and Abbeel (2017) did not consider the IB complexity measure that we are using. Second, we do not simply show that “a complexity loss can reduce complexity” or that “these losses work as intended.” Rather, we controlled EC according to principled and cognitively-motivated metrics and showed tangible benefits beyond those metrics alone: training for informativeness improved **convergence rates** (Figs 4a, 9, 10), and penalizing complexity induced more **human-like communication** (Table 1, Fig 5).
> >
> > As for Chaabouni et al’s (2021) work, it is not based only on the “natural inductive bias of neural networks” but also includes hand-crafted biases, such as the agents’ discriminative need, which indirectly affect complexity. In contrast, our approach aims to avoid such ad-hoc design choices by **training agents directly w.r.t. an objective function that human languages appear to be optimizing**. This is a fundamentally different approach to EC, and we believe that it will allow the field to explore a greater range of emergent human-like communication systems.
> >
> > As for human-AI communication, we have demonstrated that our approach gives rise to human-like color communication, by comparison with human-generated naming data from 110 languages of the WCS. We agree that also showing improved zero-shot human understanding of self-play messages is an important research direction. However, as the reviewer notes, the paper is already packed with many new ideas and results, and therefore we decided to leave this extension for future work.
> >
> > ### IV. Key Contributions and Appendices
> > The reviewer expressed concern about relegating key contributions to appendices. As also noted by this reviewer, our paper already contains “many novel insights and experiments,” so presenting the right subset of results and analysis in the main paper is challenging.
> >
> > In our revised paper, we have relegated VQ-VIB After discussion to Appendix B (allowing us to focus on results from other methods), and included additional details and plots highlighting differences between VQ-VIB and onehot communication.

---

> > > ### Author Response · Authors · 2022-08-02
> > > **Response Part 3: Specific Questions**
> > >
> > > ### **Specific Questions**
> > >
> > > 1. VQ-VIB After: Thanks to the reviewer’s feedback, we have moved VQ-VIB After to an appendix in the revised version, and now focus on VQ-VIB Before in the main paper.
> > >
> > > 2. Particle world results (Section 5.1): We have shown that VQ-VIB outperforms onehot in the sense that it achieves >= utility for the same level of complexity (Figs 4b, 9). Following the reviewer’s request, we conducted additional experiments in the uniform environment with 200 and 500 tokens, for VQ-VIB Before and onehot, and noted those results in Section 5.1. Our results are unchanged: neither method improved significantly further when using more than 100 tokens. That is, VQ-VIB continued to outperform onehot. Furthermore, the computational cost of onehot communication increased with dimensionality (e.g., a single run of onehot using 500 tokens for 10000 episodes took 10 hours), illustrating a further advantage of VQ-VIB. Because of the duration of such trials, we were unable to perform similar hyperparameter sweeps in the 9 points environment in time for the rebuttal, but we will do so for the camera-ready submission, should this paper be accepted.
> > >
> > >     - 2.1. “Why not increase vocab size more?” Performance in our framework is not measured by utility alone, but rather by the agent’s ability to maximize utility **with bounded resources**, which is more similar to what humans do. Therefore, we are interested in the utility attained with limited vocabularies rather than unbounded vocabulary sizes.
> > >     - 2.2. “If one-hot can reach the same performance/complexity at a larger vocab size then you can't say it completely outperforms it” –– to clarify, our evaluation compares performance given the same level of complexity, even if it is attained with different vocabulary sizes. The reason is that the vocab size is a hard limit which is often not realized for lower complexities (as shown in Figs 5, 9).
> > >     - 2.3. “neither agent is reaching optimal performance” – VQ-VIB attains the best performance among the **discrete** communication methods. While the continuous model attains higher utility for any given complexity, it is not ecologically valid and does not correspond to human-like communication.
> > >     - 2.4. “if you can show that VQ-VIB outperforms one-hot accuracy/complexity for any one-hot hyperparameters, then your method becomes much more interesting and your results are stronger” – in our initial submission, we varied the number of communication tokens; based on the reviewer’s suggestions, we conducted further experiments with 200 and 500 tokens. These new trials corroborated existing findings. If the reviewer has other particular hyperparameters in mind that are worth considering, we welcome their suggestions.
> > >
> > > 3. “Why does 4b show unconverged protocols?” We believe communication has converged in those results. As shown in Figure 4a, communication typically converged after 3000 episodes, and 4b shows results generated from episodes 3000 to 10000. Perhaps we have misunderstood the reviewer, and we would appreciate any clarification.
> > >
> > >
> > > 4. Section 5.2. and REINFORCE: Following the reviewer’s suggestion, we are conducting REINFORCE experiments in the color reference game. However, REINFORCE is known to have slow convergence rates and thus we were unable to complete this experiment within the 1 week rebuttal period. We will include it in the final version, if accepted, and we are confident that this will not affect our main results and conclusions because (a) with gumbel softmax (GS) we already achieve near optimal communicative efficiency, and (b) given our direct method for controlling complexity, we covered a greater span in EC complexity than what Chaabouni et al. observed when training with either REINFORCE or GS. In addition, we would like to highlight that our particle-world environment results are generated using **MADDPG**, a policy-gradient algorithm, like traditional REINFORCE.
> > >
> > > 5. “Figure 5a is suspicious”. To address this concern, we have extended our analysis to 1024 tokens, in addition to 330 tokens. A zoomed-in version of the results (with 1024 tokens) is now included in Appendix D, Fig 14, and we verified that similar trends exist in the zoomed-in version of Fig 5a  (corresponding to 330 tokens). Increasing the number of tokens did not change results, which is expected given that there are 330 WCS color chips (see Gilad-Bachrach et al. 2003 for a mathematical analysis of the effect of vocabulary size). In addition, we note that, while both onehot and VQ-VIB learn efficient communication, VQ-VIB achieves greater utility for the same complexity than onehot (as illustrated in Fig 6, which we have added to the revised version of the paper to better highlight this result).

---

> > > > ### Author Response · Authors · 2022-08-02
> > > > **Response Part 4: Citations and Clarifications**
> > > >
> > > > ### **Citation suggestions**
> > > > We already cited Lin et al.’s autoencoder work, and have clarified in the revision the distinction between our findings and approaches (Sections 2 and 5.1). We have also added the reference to Maddison et al. 2016.
> > > >
> > > > ### **Clarifications**
> > > >
> > > > - “Is the entropy regularization done in place of or on top of that loss?” — We assume the reviewer is referring to complexity equivalence classes loss from section 3.2.4. This loss is “on top of” the complexity loss: the normal complexity loss is the main training signal, and the entropy regularization is only intended to break ties between complexity equivalence classes.
> > > >
> > > > - “Is the entropy regularization in one-hot equivalent to regular entropy regularization done in previous RL works (e.g. Mnih et al 2016, Lazaridou et al 2018) or is it the opposite? Are you penalizing entropy or encouraging entropy?” — In contrast to prior work, we are considering the mutual information between inputs and signals as a measure of complexity, which agents aim to minimize. Since mutual information is the difference between the unconditional and conditional entropy, minimizing it amounts to minimizing the unconditional entropy while maximizing the conditional. Several earlier works, including those the reviewer cited, encourage the unconditional entropy of signals to prevent premature convergence. In our framework, this appears not to be necessary due to the informativeness pressure encouraging meaningful communication.
> > > >
> > > > - “VQ-VIB learns more complex communication” — This is what we intended to state. Even with no informativeness loss, VQ-VIB  agents learned more complex and informative communication than onehot agents, which enabled us to create a greater range of communication systems when annealing the complexity loss.

---

> > > > > ### Author Response · Authors · 2022-08-04
> > > > > **REINFORCE Results Included**
> > > > >
> > > > > As noted in our earlier responses, we had been unable to finish the suggested REINFORCE experiments in time for the initial rebuttal, but our experiments have now concluded. Results from using REINFORCE in the color reference game are included in Appendix D and highlighted in Figure 15.
> > > > >
> > > > > The same trends already present in earlier findings held true in these experiments. Onehot and VQ-VIB learned efficient communication, and VQ-VIB achieved greater utility, for the same complexity, than onehot. The reviewer correctly noted that Chaabouni et al. had found that REINFORCE typically learned lower-complexity communication than gumbel softmax. We overcame this challenge via our informativeness and complexity losses, which allowed us to directly control these measures. Thus, we believe that these findings demonstrate that our framework (for utility, complexity, and informativeness) and technique (VQ-VIB) generalize to even more settings than we had first presented.

---

> > > > > ### Comment · Reviewer_kucm · 2022-08-08
> > > > > **re: response part 4**
> > > > >
> > > > > thank you for clarifying your contribution in the context of previous work. I believe the reference to the failure-mode of Lin et al makes your contribution more logical in that context.
> > > > >
> > > > > clarifications
> > > > > - thank you, that wasn't clear to me in the original work
> > > > > - that makes sense, it may be worth mentioning in an appendix but is not necessary
> > > > > - I see, this is an interesting point that I believe was unexplored. I would add a mention somewhere that VQ-VIB learns a more complex protocol which is better when used with the complexity loss

---

> > > > > > ### Author Response · Authors · 2022-08-09
> > > > > > **Minor clarifications in revision**
> > > > > >
> > > > > > We are glad that the reviewer found our clarifications and comparison to Lin et al.’s failure mode useful. We have expanded upon the complexity measure of onehot in Appendix A. Lastly, we already discuss the complexity of VQ-VIB vs. onehot in Appendix D, and our latest revision notes how it is an interesting direction for future research.

---

> > > > ### Comment · Reviewer_kucm · 2022-08-08
> > > > **re: response part 3**
> > > >
> > > > 1. thank you
> > > > 2.1 this isn't a great argument when you are arbitrarily setting the vocabulary size. how did you decide what the "correct" limit for a vocabulary is?
> > > > 2.2, 2.3, 2.4. these results are very interesting and I believe they strengthen your case that VQ-VIB is an interesting algorithm whose inductive bias can outperform REINFORCE/GS for your particular metrics
> > > >
> > > > 3. this is something that is very confusing and I would appreciate a response if time allows.
> > > > by "converged", I mean that results have reached some sort of local minima and they do not change from there one. In most cooperative EC work, the protocol is relatively stable after convergence. Since you are training and plotting after convergence, does that mean that your methods protocol changes after "convergence". If so, do you have an explanation why this is?
> > > >
> > > > 4. that is fine
> > > >
> > > > 5. thank you for the zoomed-in plots in the appendix, they are much clearer to understand. as well, I believe figure 6 is a good representation of the point you're trying to make

---

> > > > > ### Author Response · Authors · 2022-08-09
> > > > > **Clarification on vocabulary size**
> > > > >
> > > > > 1. “2.1 this isn't a great argument when you are arbitrarily setting the vocabulary size. how did you decide what the "correct" limit for a vocabulary is?”
> > > > >
> > > > > The idea in our approach is to set the vocabulary size large enough so that agents could in principle use it to cover the entire input space, and then let the complexity loss determine the effective vocab size (the distinction between the codebook size and the effective codebook size in IB has been studied before, e.g., in Zaslavsky & Tishby 2019). For finite domains, like the 9-points and color domains, having a unique token for each element in the domain is enough to be able to represent a maximally accurate communication system, thus vocab sizes of 9 or 330 respectively are in principle sufficient (indeed, our results show that larger vocabularies don’t change much). In the uniform domain we considered vocab sizes that were shown to work well in prior work (Tucker et al., 2021). As you can see in Figs 5 and 12, varying the complexity penalty leads to systems with varying effective vocab sizes, i.e., agents use fewer distinct tokens even though the hard limit on the number of tokens they can use has not changed. Intuitively, you can think of our complexity term as a soft constraint on the bandwidth agents allocate for communication, which can be gradually adapted to changing communicative needs by annealing $\lambda_C$.
> > > > >
> > > > > “2.2, 2.3, 2.4. these results are very interesting and I believe they strengthen your case that VQ-VIB is an interesting algorithm whose inductive bias can outperform REINFORCE/GS for your particular metrics”
> > > > >
> > > > > thank you so much for noting this!
> > > > >
> > > > >
> > > > > 2. Our method is based on deterministic annealing, a standard technique in non-convex optimization. We use the first 3,000 steps to train agents to converge to a high-complexity (low $\lambda_C$) and high-utility communication. Then, by gradually increasing $\lambda_C$, we aim to track the local optima along the Pareto frontier with varying complexities. We only report results after these 3,000 steps and in 1,000 intervals, which is why we argue that the communication systems we report are stable solutions (as in deterministic annealing). However, each system is a stable solution for a different value of $\lambda_C$, which is why they have different complexities and categorize the space in different ways. Indeed, introducing deterministic annealing into cooperative EC is a novel idea that we present in this paper, motivated by the work of Zaslavsky et al. which linked annealing with language evolution.

---

> > > ### Comment · Reviewer_kucm · 2022-08-08
> > > **re: response part 2**
> > >
> > > Mordatch and Abbeel (2017) as well as many other works have used entropy regularization and complexity regularization (M+A penalize large vocabulary sizes). This is not directly related to IB but very directly related to the auxiliary losses you propose. Demonstrating informativeness leads to improved convergence rates has also been explored by AEComm. I do find your work to be complimentary to those but would like to point out the novelty claims should acknowledge that previous works have also used losses similar to yours with similar underlying principles (though explicitly using IB is novel).
> > >
> > > From my perspective, Chaabouni et al use fewer manually-tuned hyperparameters than your work (if I am not mistaken) and thus is less handcrafted. In contrast, your work uses explicit losses that directly reflect the metrics you care about and you manually tune those hyperparameters. For this reason, the WCS results are not as exciting to me. If you set $\lambda_I = 0$ and repeated the WCS experiments, that would be equivalent to Chaabouni et al and would indeed be a measure of REINFORCE vs VQ-VIB inductive biases. This isn't an experiment I would demand so late in the rebuttal period, but just a perspective for future reference.
> > >
> > > Thank you for moving VQ-VIB after to the discussion, it makes the paper more streamlined and I believe does a better job of focusing on your most impressive results

---

> > > > ### Author Response · Authors · 2022-08-09
> > > > **Clarification on hyperparameter tuning vs. annealing**
> > > >
> > > > > I do find your work to be complimentary to those but would like to point out the novelty claims should acknowledge that previous works have also used losses similar to yours with similar underlying principles (though explicitly using IB is novel).
> > > >
> > > > We agree that our work is complementary to prior work, and we aimed to acknowledge the relation between our losses and prior work in Section 2 (e.g., we’ve written that “Numerous works simplify communication… these methods all correspond to limiting complexity,” “Wang et al. explicitly limit complexity,” “for example, Lin et al. use an autoencoding loss… tightly related to notions of informativeness”, and “... emergent communication literature has begun to rediscover the importance of complexity and informativeness in communication...”). Following this reviewer’s comment, we have now adjusted Section 2 to acknowledge this point more clearly.
> > > >
> > > > > From my perspective, Chaabouni et al use fewer manually-tuned hyperparameters than your work (if I am not mistaken) and thus is less handcrafted. In contrast, your work uses explicit losses that directly reflect the metrics you care about and you manually tune those hyperparameters. For this reason, the WCS results are not as exciting to me.
> > > >
> > > > We do not manually tune the $\lambda$ hyperparameters, but rather gradually anneal them in order to explore the space of communication systems spanned by these (soft) constraints, much like annealing a temperature parameter. It has been argued that this annealing process may drive language evolution (Zaslavsky et al., 2018), allowing populations to gradually adjust their communication systems to changing communicative pressures. Thus, our motivation for varying these parameters is to simulate agent adaptation and language change, rather than to fit to the WCS dataset as in Chaabouni et al. 2021. Our comparison with the WCS data suggests that this annealing process, in addition to capturing how actual color naming systems evolve, may also guide the evolution of human-like communication systems in artificial agents.
> > > >
> > > > In addition, we believe that our approach is more theoretically justified because, based on extensive existing research on human languages, humans are more likely to be optimizing for a complexity-informativeness tradeoff than for constraints like Chaabouni et al.’s discriminative need (intuitively, it is unlikely that speakers of Ifugao, a language with 1.5 bits of complexity, never need to discriminate between two similar colors).
> > > >
> > > > > If you set $\lambda_I = 0$ and repeated the WCS experiments, that would be equivalent to Chaabouni et al and would indeed be a measure of REINFORCE vs VQ-VIB inductive biases. This isn't an experiment I would demand so late in the rebuttal period, but just a perspective for future reference.
> > > >
> > > > We have actually done this experiment already and refer to results from running REINFORCE with $\lambda_I = 0$ in Appendix D.2; we found that training with onehot communication was unstable and often collapsed to no meaningful communication (complexity = 0). We omitted explicit discussion of VQ-VIB with $\lambda_I = 0$ for brevity besides a general note that “VQ-VIB agents typically converged to higher informativeness (and higher complexity) communication than onehot when complexity was not penalized.” Given the reviewer’s request for more specific data, we analyzed our saved data from trials with $\lambda_I = 0$ and found that complexity for VQ-VIB converged between 1.4 to 1.9 bits over 5 random trials. This exactly demonstrates the differences in inductive biases between VQ-VIB and onehot.

---

> > ### Comment · Reviewer_kucm · 2022-08-08
> > **Re: response part 1**
> >
> > thank you for clarifying the framing. I still find that you are proposing two ideas (utility, complexity, informativeness - VQ-VIB) to be two mostly-separate approaches to the idea of incorporating ideas from IB into emergent communication. Indeed you could likely do the same two extra losses (complexity, informativeness) for a standard REINFORCE gradient estimator. Perhaps, there is a reason why VQ-VIB is the only method that can perfectly work with these two extra losses in ways that others can't? If so, it could be useful to spell out explicitly. But I think figure 9 demonstrates at least that informativeness loss is a general idea that can be applied to many algorithms (this is a good thing!)
> >
> > thank you for the additional experiments (I will address them in later comments). I acknowledge that your work is not aiming to beat some sort of benchmark but emergent communication is an inherently empirical field. You don't necessarily need to demonstrate better results but something novel compared to previous approaches. Based on your response, I will take the idea of greater utility for a similar complexity to be that (you could also argue it being a more human inductive bias but then I would prefer to see more experiments related to that, though WCS is a good start).

---

> > > ### Author Response · Authors · 2022-08-09
> > > **Thanks; further discussion of advantages of VQ-VIB**
> > >
> > > Thank you for the encouraging response! We’re very happy that our earlier responses helped clarify the framing and contributions of this work.
> > >
> > > > Indeed you could likely do the same two extra losses (complexity, informativeness) for a standard REINFORCE gradient estimator. Perhaps, there is a reason why VQ-VIB is the only method that can perfectly work with these two extra losses in ways that others can't? If so, it could be useful to spell out explicitly. But I think figure 9 demonstrates at least that informativeness loss is a general idea that can be applied to many algorithms (this is a good thing!)
> > >
> > > In contrast to other architectures, VQ-VIB is more similar to VIB in that it explicitly learns an encoder which is formulated as a Gaussian distribution over a latent space. But in contrast to VIB, VQ-VIB also learns an underlying symbolic structure by simultaneously learning a discretization of the latent space into prototypes. In that sense, VQ-VIB enjoys both worlds: it has a symbolic structure, like Proto, and to some extent like onehot (although onehot doesn’t have prototype embeddings), but it is also better suited for learning an IB encoder. Thank you so much for pushing on this point. We agree that this clarification is important. We also agree that the fact that we’ve shown that other architectures may also benefit from our informativeness loss is noteworthy on its own, as a secondary contribution of our work. We have added these important insights to the paper in Section 6.

---

### Author Response · Authors · 2022-08-02
**Summary of individual comments**

We thank all the reviewers for their helpful and thoughtful comments. We have uploaded a revised submission to address the reviewers’ suggestions, and in our detailed responses to each reviewer we address all the concerns they raised. Below is a summary of the main changes in the revised submission.

### Summary of revisions:

1. Following reviewer kucm’s comments, we (i) included additional experiments to extend the range of lexicon sizes (added 200 and 500 tokens in the uniform env, 1024 tokens in the color reference game); (ii) added Fig 6 to highlight the advantage of VQ-VIB over onehot; and (iii) added a zoomed-in version of the color reference game results (Appendix D, Fig 14). We also conducted additional experiments with REINFORCE, as suggested by the reviewer (Appendix D, Fig 15). **All new experiment results requested by the reviewer corroborate trends identified in the initial submission.**
2. Both reviewers kucm and 3YrT note that VQ-VIB After is not as interesting as VQ-VIB Before, and reviewer kucm also notes that many results are discussed too briefly in the main text. We have therefore moved VQ-VIB After to an appendix, and instead focused more thoroughly on the key results in the main paper.
3. Reviewer 3YrT has raised concerns about our definition of complexity and pointed to the debate in the literature on the notion of linguistic complexity. In our detailed response to the reviewer, and in the paper, we explain the theoretical and empirical justification for our particular choice of complexity, especially in the context of **lexical semantics**. In addition, following the reviewer’s comments, we have added a note to Section 2 about this debate in the literature, including the relevant references suggested by the reviewer.
4. Following reviewer 1FYj, we extended our description of the Proto architecture in Section 4.2.

---

### Meta-Review · Area_Chair_LbDe · 2022-08-27

**Recommendation:** Accept
**Confidence:** Less certain

**Metareview:**

From the ratings alone this paper appears borderline leaning towards acceptance, however, I want to highlight to the authors that in discussion with reviewers and my own reading of the paper there are aspects that shifted this even closer to the decision boundary. In the end, my own conflicted views of the work and the lack of further discussion from the more negative reviewer led to a recommendation of accept.

I'll briefly review some of the strengths and weakness in the latest revision as I see them.

+ The work truly studies the effect of controlling multiple objectives of communication in the emergent communication setting and how these affect trade-offs in complexity, informativeness*, and utility. This scientific approach to the experimental work is in my view a clear strength of the work.

+ There is a clear motivation for the choice of objectives based upon existing work in related fields. Although I have reservations about how well the realization, building on the Zaslavsky et. al.'s work and within emergent communication is something I think will benefit the community.

+ The writing itself is clear and easy to read. Figures in the main text and appendix were informative and quite interesting.

- Important details, especially around the math and experimental setup, are lacking. Some of thing examples that bothered me were: ambiguity on definition of terms in the objective (all three terms could be stated more precisely but for U(X, Y) we are not told what Y is and for I(X, C) it is not clear if this is coming from 3.2.2 or 3.2.4); lacking clarity around gradient flow (passing gradients back to the sender in this type of setup should be stated very clearly and upfront).

- Connection between I(X, C) and complexity. As I was reading I interpreted this to be (as in 3.2.2) "the KL divergence of μ(x) and σ(x) from a unit Gaussian", and found this is to be a very strange choice (not as an objective but as a measure of the complexity of the language). This is, I believe, a different concern than what was raised by one of the reviewers. The issue I saw there was that the codebook need not be uniformly distributed in the continuous space, and therefore the implications on message complexity of a particular variance in the continuous latent space may be inconsistent. For some regions of latent space a unit variance could imply a single message with high probability, while other areas of the latent space may be more closely clustered causing the same unit variance to be nearly uniform over several different messages. Moving on to 3.2.4, we see (perhaps) evidence that the authors encountered the repercussions of this choice "simply penalizing this term in training was insufficient to train VQ-VIB agents to use fewer unique discrete embedding". The motivation is great, but I strongly suspect that there are better ways of realizing it and that this particular choice is not capturing what the author(s) intended.

- Limited novelty of the method. The combination of VIB and VQ-VAE seemed like something that would have already been published, since regularizing the prior of the latent distribution is such a well used (maybe even well understood) method, but I looked and must grant that this does seem to be a novel combination. That said, it is not so novel that it would be able to stand on its own without the specific setting itself and connection with models of complexity in human languages to support it.

- (Minor) The author(s) made several additional changes to address reviewer concerns around prior work and putting their design choices into context, but there remain some issues in this space. I found the discussion of related work to be fairly shallow and at times even dismissive. It reads as though the work was undertaken entirely devoid of consideration for related work except for that which directly motivated the approach, and then was added defensively but without making real connections between this work and others. I mark this as minor because it is unfortunately somewhat common and because it is a more subjective evaluation.

I hope this and the other reviews will help author(s) to understand how the work may be experienced by readers and potentially make further refinements. Overall, despite limitations, I do believe this work will be of interest to researchers in emergent communication and potentially more broadly due to common underlying questions around trading off complexity and other primary learning objectives.


**Award:**

No

---

### Decision · Program_Chairs · 2022-09-14

Accept